



# Dynamics of Large Pelagic Ice Crystals in an Antarctic Ice Shelf Water Plume Flowing Beneath Land-Fast Sea Ice

Craig Stevens[1,2], Natalie Robinson[1], Gabby O'Connor[2] and Brett Grant[1]

[1]National Institute of Water and Atmospheric Research, Greta Point, Wellington, New Zealand.
[2]University of Auckland, New Zealand.

*Correspondence to*: Craig Stevens *craig.stevens@niwa.co.nz

**Abstract.** Observations of boundary-layer processes and ice crystal behaviour in an outflow region from the Ross/McMurdo Ice Shelves are presented. From a fast ice field camp, we captured the kinematics of free-floating relatively large (many 10s of mm in scale) ice crystals that were advecting as well as aggregating in a depositional layer on the sea ice underside (SIPL, sub-ice platelet layer). Simultaneously, we measured the background oceanic temperature, salinity, currents and turbulence structure. At the camp location the total water depth was 536 m, with the uppermost 50 m being in-situ super-cooled. Tidal flow speeds had an amplitude of around 0.1 m s$^{-1}$ and the resulting under-ice boundary layer sustained turbulent dissipation rates as large as $\varepsilon = 10^{-6}$ W kg$^{-1}$. Acoustic sampling (200 kHz) identified three classes of backscatter (1) large individual highly mobile targets, (2) echoes from large, individually identifiable suspended crystals and (3) a varying background, presumably of very small (frazil) crystals. This second class of backscatter was associated with crystal sizes far larger than typical, certainly larger than anything normally described as frazil, and some individuals at least were depositing close to "fully grown". Measurement indicated crystal scales of the range 30-80 mm. The existence and settlement of this scale of crystal has implications for understanding SIPL evolution and the processes controlling the fate of Ice Shelf Water.

## 1 Introduction

Regional variability in Antarctic sea ice is a major issue for climate prediction, challenging models (Ludescher et al., 2019) and confounding communication of key issues to stakeholders and decision-makers. With anthropogenically-induced warming oceans penetrating farther south, increased ice shelf basal melting is expected (Rignot et al., 2013; Kusahara 2020). Relatively warm ocean water penetrates the interior of ice shelf cavities and induces melting on the ice underside. One driver of sea ice variability is the feed-back effect of meltwater exiting major ice shelf cavities (e.g. Holland et al., 2007; Langhorne et al., 2015). The resulting water is at the local freezing point temperature, as dictated by pressure and salinity. This water mixes with the ambient ocean resulting in a fresher, cold seawater





plume that seeks out the fastest upward flow path on the shelf underside subject to the Coriolis force and basal slope (MacAyeal 1985; Jenkins and Bombusch 1995; Smedsrud and Jenkins 2005; Stevens et al., 2020). These plumes will grow with sustained melting and/or decay with re-freezing removing their thermal deficit. If the plume persists sufficiently to reach the ice shelf edge it flows out beneath the neighbouring sea ice margin (Fer et al., 2012; Langhorne et al., 2015). At this point the basal slope driver of flow ceases and the persistence of the supercool plume is controlled by initial buoyancy, growth of new ice, topography and mixing (Hughes et al., 2014). This step represents a critical phase in the passage of basal melt water where the greatest changes in drivers and environment are all located. This supercool water drives sea ice growth by absorbing heat into the stratified upper ocean and facilitates the generation and growth of ice crystals (Robinson et al., 2014; McPhee et al., 2016; Hoppmann et al., 2020).

There is evidence that in some settings this ice formation occurs as buoyant crystals in the water column (Hoppmann et al., 2015; 2020; Frazer et al., 2020). If these crystals grow slowly, remaining sufficiently small that viscosity dominates, then they are mainly passively advected. Typically, this is the scale (<1 mm) at which crystals are thought to exist. However, if they grow sufficiently large whilst suspended, then buoyancy-driven thin disk mechanics must dominate their trajectory (Jordan et al., 2015). The extent of a plume has been modelled using schemes that develop and transport crystals, again with a focus on mm-scale crystals (e.g. Holland et al., 2007; Hughes et al., 2014), or if not, then in a "bed-load" and so essentially a part of the ice-ocean interface (Robinson et al., 2014). Sampling challenges make it difficult to build up spatial appreciation of the crystal metrics and growth rate in the SIPL but one correlation that emerges is that SIPL thickness and supercooled seawater are co-located (Langhorne et al., 2015; Brett et al., 2020).

By following the ice shelf water plume as it evolves in space and time, it is possible to look at ice growth and thermal relief (e.g. Smedsrud and Jenkins 2005; Hughes et al., 2014). A recent review by Hewitt (2020) identifies issues like crystal growth, the role of sediments and the limited availability of observations as being key issues for the advancement of understanding at the ice shelf scale. At the larger regional to global scale the challenges lie more with sea ice production and water mass formation as coupled models seek to combine the ocean, atmosphere and ice structure (e.g. Roach et al., 2018; Richter et al., 2020; Moorman et al., 2020). Uniformly these studies identify the need for more observations both at the process, and monitoring, scales.

A decade-long sequence of sea ice camps in the McMurdo Sound region (Robinson et al., 2020) have revealed that these platelets form a coherent layer on the underside of sea ice (a sub-ice platelet layer, SIPL Hunkeler et al., 2015; Wongpan et al., 2015; Hoppmann et al., 2020) – into which they are eventually incorporated (Smith et al., 2001; Langhorne et al., 2015). The McMurdo Ice Shelf, a small ice shelf that sits between the Ross Ice Shelf (the largest ice shelf on the planet by area) and McMurdo Sound. It includes a region called "the Dirty Ice" because of the substantial rock debris visible on the





shelf surface and has been described as "perhaps the strangest ice shelf in the world" (Debenham, 1965; Atkins and Dunbar 2009). This material is partly marine in origin, as sediment entrained into the growing marine SIPL on the shelf-underside finds its way to the surface (Campbell and Claridge 2003). This, in itself, is evidence of supercool oceanic conditions.

Hoppmann et al. (2020) review our present understanding of Antarctic platelet ice and makes it clear the topic is still in a discovery phase – partly due to the challenges of making comprehensive observations. At the same time, modelling approaches have needed to advance – creating a tension. It is likely that the deposition and formation of ice crystals at a range of scales influences interfacial momentum transfer, sea ice composition and strength as well as ecological habitat throughout localised

parts of Antarctic coastal waters. While geophysical boundary-layers are well understood, a number of questions arise around unique aspects of the present situation and provide a focus for the present study. (1) Is there evidence of large pelagic crystals? (2) Is there a relationship between crystal behaviour and the turbulent under-ice boundary-layer structure? (3) Does sediment from the Dirty Ice influence the mechanics? (4) What are the large-scale implications of such finescale mechanics?

## 2.     Methods

### 2.1 Location and Camp

     The "K131 sea ice camp" (Antarctica New Zealand logistics event designation, Stevens et al., 2018) was deployed on 2.3 m thick sea ice from 21 October-4 November 2015 at a location in Southern McMurdo Sound (-77° 51.913' S   +166° 00.053' E) on an ocean depth of 536 m. At the time of

90 sampling the edge of the fast ice was ~20 km to the north. The camp location was selected based on surveys of platelet deposition thickness (1.8 m) whereby there would be sufficient platelets to be measurable (Langhorne et al., 2015), while avoiding too many platelets (i.e. SIPL too deep to easily penetrate a conductivity temperature depth - CTD - profiler through) and also avoiding the substantial tidal currents encountered further towards Haskell Strait (Figure 1a, Stevens et al., 2009; Mahoney et

al., 2011). The K131 camp consisted of modified shipping containers with cut-out floors allowing access to the ice and ocean below. A hot-water cutter was used to melt through and remove the sea ice in blocks. It was notable that upon removal of the blocks, the water which filled the hole appeared milky but that this gradually dissipated over the subsequent days. After 12 days of operations and many seal occupations of the holes, the hole water was fully flushed and very clear. We speculate that the

water in the hole was initially from the melting of the sea ice and upwards drainage from the SIPL and contained sufficient levels of sediment to be visible but that over time this was replaced with clear ocean water. With regard to the regional sampling context, the present data were collected south of where the later Frazer et al. (2020) acoustic sampling took place. Furthermore, the analysis of under ice



boundary roughness data synthesis described in Robinson et al. (2017) include some data from the same
field camp as here, but with instruments focused on the ice underside.

## 2.2 Sub-Ice Platelet Layer (SIPL)

Upon removal of the blocks of sea ice, the SIPL would remain intact beneath the ice hole, giving
an indication of the cohesive strength of the layer. Sampling activity would then penetrate through this
SIPL by first lowering a weight to create a hole. Hoppmann et al. (2020) describes the semantics of
platelets and frazil. For present purposes we consider them to be ends of a spectrum of the same
physical crystal evolutionary process. Crystals were extracted from the SIPL by reaching down with a
porous scoop. They would be recovered either as individuals or in clumps. In many instances once
extracted the clumps would collapse with very little encouragement indicating only weak interlocking
crystal growth. This in part indicates the role of buoyancy aiding maintenance of the SIPL but also
suggests that the SIPL is not always a tightly interlocked matrix of crystals. The dimensions of each
crystal were then quantified by manual measurement (done immediately to avoid significant melting or
growth).  It is possible that these represent a distinctly different set of sizes to the suspended pelagic
crystals, but large crystals were definitely observed in video rising. It is likely however that this
approach is biased towards larger crystals (> 5 mm diameter). Crystal size data from a nearby camp
from the following year (77.8183°S 165.4059°E, November 2016 – see Robinson et al., 2020) are also
included.

## 2.3 Video and echosounder records

While qualitatively very useful, video/visual observations are challenging to interpret
quantitatively because of light variability, lens distortion and reflection/viewing angle effects.
Additionally, it was difficult to see individual crystals arrive and settle as the light would saturate the
camera.  Penrose et al. (1994) describe how acoustics provide a more consistent picture of crystal
behaviour.  We used a Simrad EK60 200 kHz echo sounder recording acoustic backscatter at 1 Hz with
4 cm vertical resolution over a sampling cone that is 7 degrees across so that at a depth of 25 m the cone
is three m wide.  The cone has side-lobes in the upper 5 m of the water column that are as wide as 30
deg.  The beam-width is not particularly critical so long as it is wide-enough that scatterers register
some rise component (vertical resolution).  With horizontal flows of around 0.05 m s$^{-1}$ this means that a
reflector would stay in the beam for a maximum of nearly two minutes at 5 m with a 30-degree cone.
Any crystal rise (or fall) is seen as an oblique trace in the backscatter timeseries field, so that each
streak in the echogram corresponds to a single crystal. An analysis was developed to quantify local
vertical motion over the 270 s from scan to scan. Each acoustic profile was first scaled to remove depth-
attenuation of acoustic strength. Then the data were stepped through in time, breaking each profile into
vertical segments. For each vertical segment, the next profile in time was searched for where this

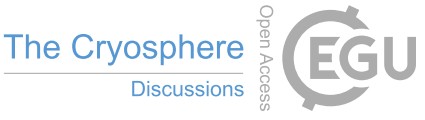

segment had the best match in terms of intensity distribution. This allowed synthesis of a distribution of rise speeds. Despite superficial appearances, the angled trajectories relate to the crystal rise speed and

are unconnected to the tidal advection. Horizontal flow instead controls the persistence of individual reflectors as it is responsible for moving crystals in and out of the acoustic beam.

## 2.4 Profiling instruments

Water column temperature and salinity were recorded using a SBE19+ CTD profiler that was calibrated pre- and post-experiment resulting in an accuracy of ~5 mK. Care was taken to avoid ice growth on the

CTD sensors, and also to thermally equilibrate prior to profiling (Robinson et al., 2020). The example used here is a down-cast from an instrument that was held for one hour at 200 m prior to bringing up into the hole very briefly and then profiling downwards.

In addition to the CTD profiler, a Rockland VMP 250 microstructure profiler was deployed to quantify turbulent mixing by capturing fine and microstructure scale variability. This loose-tethered

profiler falls at ~0.65 m s$^{-1}$, recording velocity shear and temperature at the microscale and fine-scale temperature and conductivity along with some other properties. The microscale shear enables estimates of energy dissipation rate $\varepsilon$ (Wolk et al., 2002) and has previously been successfully deployed beneath Antarctic fast ice (Robertson et al., 1995; Stevens et al., 2009; Fer et al., 2012). Eighty-one microstructure profiles were recorded over a three-day period (1-3 Nov. 2015).

## 2.5 Moored instruments

Three pairs of current meters (Aanderra RCM 11) and conductivity-temperature-pressure sensors (Seabird Electronics SBE 37), both sampling at two-minute intervals, were deployed on a suspended mooring located at 77° 51.903' S 166° 00.351' E, 200 m north east of the main sampling location. The instrument pairs were deployed at 32, 82 and 374 m beneath the surface. In addition, ten SBE 56

temperature loggers sampling at 1 Hz were deployed at 5, 10, 15, 20, 43, 53, 141, 199, 257, 315 m. All the upper instruments were affected by icing issues (Robinson et al., 2020) to varying extents and only used in specific instances here.

## 3    Results

### 3.1 Background water column conditions

Water column temperature structure showed that the ocean temperatures were mostly above the pressure-dependent freezing level (Figure 2). The exception to this was the upper ~50 m which was always in situ supercooled, sometimes with a clear change in structure at the depth of transition to non-supercooled. The upper layer temperature/salinity ($S_P$=34.65 psu, $\theta_0$=-1.94 °C or for TEOS-10, $S_A$=34.82 g Kg$^{-1}$, $\Theta_0$=-1.94 °C) indicates Ice Shelf Water (ISW) conditions. The potential temperature





comes close to freezing again at 80 m but beneath this, even though temperatures remained below $\theta_0$=-1.91 ℃, the water was not in situ supercooled. The relatively homogeneous upper layer salinity (34.65 psu) gives away to a quasi-linear increase below 100 m.

The deepest current meter provided the best quality current speed results (Figure 3a,b). The upper current meter did work for a few days at the beginning and was sufficient to show that the upper speeds were between 50 and 100% faster than the deep flows, at least in the conditions at the time. This is likely the result of the buoyantly forced ice shelf water plume. The experiment commenced near the end of spring tides followed by neap tide around DOY 297-299 and then the following spring tides peaked around DOY 304.5. The tidal speeds are clearly apparent with an amplitude of 0.1 m s$^{-1}$. However, they are not the only advective process, as a period of flow near the start of the experiment showed speeds in excess of 0.1 m s$^{-1}$ and consistently moving towards the north (i.e. away from the ice shelf). Temperature and salinity were consistent over the period but still responding to the tide at times.

### 3.2 Ice crystals and backscatter

The majority of the measured crystal dimensions ranged from 5 mm through to 200 mm (Figure 4). The average plan-view dimension in 2015 was 93 mm (with a slightly larger 101 mm equivalent measured the following year). Thicknesses were 2-10 mm, with the thicker ones clearly multi-layered (thickness was not measured in the subsequent data set). The ice crystals (Figure 4a) were often larger than 100 mm in apparent scale (Figure 4c) and while these are found in the well-defined SIPL beneath the sea ice, it became clear that there was a constant supply of crystals from depth, some of which were already of large scale.

There appeared to be two types of behaviour from visual video observations (Figure 5). The first was seen at around 10 m depth platelet crystals were being advected horizontally but with some randomness to direction and not always with an obvious upwards component. The second type was seen in imagery from just beneath the ice (~ 1 m) which showed a more ordered region of suspended crystals, especially at slack water. The smaller ones drifted slowly horizontally coherently while the larger individuals were occasionally and independently seen rising into the SIPL.

The 200 kHz echosounder provided a new perspective on the presence and behaviour of these suspended crystals as various acoustic backscatter conditions were observed over the nine days of sampling (Figure 6). These conditions included large individual biological agents observed against a slowly varying background (Figure 6a) on occasionally with a more rapidly varying background (Figure 6b). It was also common to observe a varying intensity in the background field (Figure 6c), but individual target streaks would persist through the signal variation. The target strength was not a reliable separator however, as there would occasionally be strong scatterers that simply rose and entered the SIPL. Interpretation of video suggests these are relatively large crystals.



The analysis of crystal rise speed (Figure 7) found only a moderate bias to upwards flows and
quite small velocities (mostly less than 1 cm s$^{-1}$). With the larger crystals being around 7 cm in
diameter and rise speeds of the order of 1 cm s$^{-1}$, this implies a Reynolds number
Re=0.01x0.07/10$^{-6}$=700. Instability behind a buoyant disk commences well below this at Re=~60
(Natarajan and Acrivos, 1993). However, a significant proportion of crystal speeds are downwards –
i.e. against buoyancy (positive velocity as shown in Figure 6b). This suggests that the upper water
column is some combination of (1) internal wave motion that is as likely to be downward as it is
upwards, (2) isotropic shear-driven mixing and/or (3) convective instability whereby the crystals are
entrained into drainage plumes from the SIPL.

The vertically integrated acoustic backscatter doesn't show any obvious consistent correlation
with the velocity and scalar properties. There are some exceptions to this though. For example, near the
end of day 298 (peak 2 in Figure 3) at which point we see shift from uni-directional flow through to a
tidal oscillation and this comes at the end of a period of dropping temperatures (although only dropping
by 30 mK) and increasing salinity. Simultaneously we observed the rise period to the highest
backscatter (which is log-scale db). There is another instance where speed peaks correspond to changes
in temperature, salinity and backscatter (peak 5). Near the end of the experiment at peak 7 again a
velocity peak coincided with a change in backscatter, flow direction and salinity suggesting an entirely
new water mass was moving by. It is noteworthy that the 3-4-day trends are comparable between
temperature and acoustic backscatter as the pre day 299 conditions give way to a decline in backscatter
while temperatures rise steadily over days 299-302. After day 302 the trends in both temperature and
backscatter remain flat. These periods also correspond to changes in the tidal structure with
unidirectional flow prior to day 298 then transitioning (days 299-302) to steady back and forth tides
(post day 301).

When considering the frequency structure (Figure 8), the timeseries are not long enough to
enable analysis to extend much lower than the diurnal tidal frequencies. The current meter spectrum is
dominated by the tide with almost one third of the spectrum reaching an apparent noise-floor. The
temperature spectrum (from the sensor within the supercooled upper ocean) broadly conforms to three
sections, the upper and lower of which both follow a $f^{-5/3}$ slope consistent with an isotropic turbulent
cascade. However, this is interrupted with a band between 20-80 cpd (marker c in Figure 8 i.e. 20-70
min). The echo sounder spectrum can be broken into several components. Between the tidal frequency
and 100 cpd (marker a i.e. 15 min) the slope is less than unity. Beyond this the slope is much steeper,
with a local peak at 60 cpd (marker b i.e. 24 min). This could either indicate there is enhanced energy
in this range, or that the vertical averaging serves to dampen higher frequencies. Considering a
convective velocity scale of say 5 mm s$^{-1}$ (as indicated by ε) cycling over a surface layer of 50 m
suggests a turnover time of around 3 hours (8 cpd). This suggests the variability we are seeing is
happening relatively quickly compared to the mechanical operation of the upper layer. Conversely, the





variations in background signal amplitude seen in Figure 6b and c occur around 10 s intervals so around
9000 cpd. These sit well into the high frequency tail of the spectrum (Figure 8). Another scale of
variability that will be apparent at least close to the surface is the SIPL underside has around a 2-5 m
undulation which coupled with a 0.1 m s⁻¹ implies a 20 s variation. This too falls to the right in the high
frequency content.

### 3.3 Boundary-layer turbulence

The VMP profiles revealed mostly good quality turbulent spectra (Figure 9) allowing for reliable
estimation of ε which provides insight into the dynamics of the vertical structure of suspended crystals
(Figure 10). The time-averaged temperature and salinity profiles (over around 48 hours) supported the
structure from the CTD profile of a homogeneous water column. The averages revealed an apparent
warm, salty layer in the 5 m just beneath the SIPL. Note this layer is not apparent in all individual
realisations. The temperature effect on stability is almost passive in this temperature range and so does
not compensate for the observed salinity increase and so the upper 15 m, on-average, is weakly
unstable. It is possible that the near-SIPL instability it is an effect of the ice hole, but the water in the
hole itself is stabilised with much fresher water and it is not clear how rapid response sensors would
register any false salinity readings. In addition, equivalent conditions have been observed just beneath
the SIPL elsewhere (Robinson et al., 2014). A time-averaged profile of turbulent dissipation rate ε
(Figure 10b) is dominated by the higher values but generates a smooth profile, the bulk of which is
comparable to a scaling argument for ε based on the characteristic turbulence velocity scale $u_*$ and the
distance from the boundary ($z$) so that $\varepsilon = u_*^3/(Kz)$, where $K$ is von Kármán's constant ~0.41. However,
this fit works best beneath the upper layer (i.e. deeper than 10 m).

### Discussion

### 4.1 Is there evidence of large pelagic crystals?

The measured crystal sizes represent the large structural material of the SIPL. However, it is
clear that large crystals can exist in suspension as we could see them visually and can be a source for
the SIPL. Visual observation could not see the presumably more typical <1 mm crystals (e.g. Frazer et
al., 2020). It remains uncertain as to what proportion of the large crystals extracted from the SIPL
underside had grown in situ vs directly deposited. Given that they were barely amalgamated (c.f. the
"loose" platelets of Arndt et al., 2020) it seems possible that they are recent large arrivals. On the other
hand, it seems implausible that such large crystals couple float upwards but do not generate a much
higher rise velocity.

As Hoppmann et al. (2020) explains, the understanding of the initiation of crystal formation is
not yet well established. Generally, active acoustic probing provides a useful tool to interpret suspended



particle dynamics. Away from ice shelves it is usual to relate acoustic backscatter to a combination of individual motile biological targets and a background continuum signal. In the open or coastal ocean

these usually relate to different species scales or suspended sediment. In such situations, fish and large zooplankton present as individual targets whilst the phytoplankton or sediment form a background continuum signal. The conditions in the present data suggest there is a high probability that the majority of our observed targets are pelagic platelet crystals and that the net upwards transport (Figure 7) suggests they form at depth and are not grown from surface-induced processes.

In the present sub-ice situation, the acoustic backscatter data falls into three categories – (1) large, bright individual backscatter, (ii) medium scale "less-bright" individual backscatter targets and (iii) a slowly varying background field. The Frazer et al. (2020) observations took place nearby (in a different year) and used four separate acoustic frequencies including one matching the present EK60. The nature of the ISW outflow is not consistent from year to year. In the present context they primarily

captured the background field, as the majority of their crystals were estimated to be around 1 mm in scale and any large crystals would have been removed by the processing. Video observations and target behaviour support the contention that the majority of rare very "bright" individual targets were large fauna – ranging from fish, fish schools through to seals. However, it was far more common to observe a hybrid of the 2nd and 3rd conditions whereby the sampled field consisted of many less bright but still

clearly individual signals against a coherent background field. The present situation is downstream of the Ross Ice Shelf cavity within which residence times in zero light likely to be in the range of 1-5 years (Reddy et al., 2010; Stevens et al., 2020). This reinforces the contention that the targets and continuum are ice related rather than suspended biological or sediment.

Here we refer to the large, mature crystals at depth as pelagic crystals to distinguish them from

crystals already integrated and growing in the SIPL. Laboratory and numerical work demonstrate that buoyant disks rotate so that their flat face is roughly horizontal although in some circumstances it is possible for an oblique equilibrium to exist which might be the cause of some of the horizontal motion observed here (Fabre et al., 2012). Furthermore, the behaviour of rising disks can be connected back to the initial conditions, suggesting attention be paid to the spontaneous growth from a very small nucleus

(Daly 1984; Tchoufag et al., 2014). Assuming the freshwater results of McFarlane et al. (2014) provide a minimum rise velocity (i.e. here we expect greater buoyancy different between crystal and saltwater) then their results for non-vertical disks suggests $w_c=\alpha d$ (with $\alpha=1$ s$^{-1}$, at least for diameter d<6 mm, where the one-to-one equivalence has no dynamic significance). Rees Jones and Wells (2018) use one of the faster rise rates from the same set of results which is applicable to smaller crystal sizes (e.g. none

of which conforms to the drag based estimates described in Daly, 1984) whereas Matsumura and Oshima (2015) use a fixed rise rate of 1 mm s$^{-1}$, again for small crystals.

Production of crystals at depth and its subsequent integration into sea ice is a key step in the formation of the SIPL, at least at this location (Hoppmann et al., 2015; Hunkeler et al., 2015). This has



several implications, the most important being an alternate pathway for platelet arrival and structuring
of the SIPL. If they arrive essentially mature at the ~5 cm scale, this is very different to arriving at the 1
mm scale and then growing (Dempsey et al., 2010) and the categories of backscatter described earlier
suggest both happen. The issue of orientation suggests that the arrival velocity will be slower than if
they were to rise in some other orientation with a reduced drag profile. Thus, there may be a correlation
between rise speed and packing in the SIPL. Potentially this alignment relates to the large-scale
variability seen in Figure 6c whereby large parts of the domain change backscatter but individual
scatterers are clearly seen through the transition. In other words, horizontal flow is slow, yet the
scattering still changes as crystals re-orientate themselves within the sensing volume.

The influence of pelagic crystals can be represented in larger-scale models such as Kim et al.
(2006) and Roach et al. (2018). However, it will require some local-scale mechanics. For example,
Dempsey et al. (2010) and Wongpan et al. (2015) simulate this aggregation by injecting a continuous
flux of crystals from the ocean at the same time as recording (i) the rate at which the sea ice
incorporation front moves downward and (ii) below this front but still within the SIPL, crystals continue
to grow. In their approach, they maintain different size classes for the deposited systems as opposed to
those floating up from below. This implies that there is an ability for the rising crystals to fill in the
interstices of the SIPL, reducing the void fraction. This is a point made by Dempsey et al. (2010) who
quantified the flux rate of 4 mm diameter platelets required to grow the observed sea ice to be of order
$10^6$ crystals m$^{-2}$ d$^{-1}$.

The pelagic growth to the crystals also means that the brine rejection will happen essentially
within the upper ocean layer as well at the sea ice underside. This affects the upper layer turbulence
and entrainment. In turn, this influences the persistence and fate of the ISW plume. This shifts the
buoyancy flux inherent in the energy conversion from a boundary process to what is effectively an
"internal buoyancy source".

### 4.2 Crystal behaviour in the turbulent under-ice boundary-layer

It is useful to compare vertical rise rate of crystals $w_c$ with turbulent mixing in the water column.
The present profiler-resolved dissipation rates are comparable with Fer and Widell's (2007) data from
beneath Arctic sea ice in a fjord. However, while both studies observed turbulent energy dissipation
rates in the range $\varepsilon=10^{-7}$ to $10^{-6}$ W kg$^{-1}$, their results were from a faster-moving water column. It would
appear the rougher ice underside here increases the turbulence to provide apparent matching conditions.
In the upper water column (5-10 m), the dissipation rate is an order of magnitude, or more, greater than
the value expected if we match the deeper $\varepsilon$. One hypothesis is that we are seeing brine rejection and
associated enhanced turbulence. Regardless, the $\varepsilon$ provides a dynamic context for considering how the
pelagic platelet crystals behave. The u* is a combination of convection-induced turbulence and drag-
induced stirring. Observations of boundary-layers beneath platelet ice have suggested that the drag




coefficient is a factor of 6-30 times larger than might be expected for a smooth, melting ice surface
(Robinson et al., 2017). This reinforces the apparent paradox that melting ice (the shelf basal
underside) produces re-freezing that then affects how the entire system circulates. A key knowledge
gap highlighted then is the under-shelf mixing in the basal melt layer. This will influence not only the
sub-shelf re-freezing but the amount of supercooled water being ejected into the sea ice system.

Fer et al. (2012) observed crystal-laden water emerge from beneath the Brunt Ice Shelf into an
ocean with 3 m thick fast ice. They did not observe an unstable temperature-salinity structure, but this
may be due to sampling differences when working from a ship and through a larger hole. They did
observe an increase in ε when supercooling conditions occurred, reaching around $5\times10^{-7}$ W kg$^{-1}$ when
temperatures fell to 30 mK below surface freezing. They attributed the increase in turbulence to being
due to crystal formation and rise. Conditions were comparable to that observed here with flow speeds
reaching 0.15 m s$^{-1}$. They too observed ice-related increased in 75 kHz ADCP backscatter although the
resolution is much coarser than possible with dedicated echo sounders.

Mean stratification in the present 2015 K131 work is very weak in the upper 50 m so diffusion
in this layer (thickness $h$) is estimated as $K_v \approx u_* h$ where u* is derived from the observed dissipation rate
by assuming $\varepsilon = u_*^3/h$. Given the $K_v$ model of redistribution of material in the vertical, the relative
influence of sinking (or rising) compared to the turbulent influence may be gauged using a turbulent
particle Péclet Number (O'Brien et al., 2003) $Pe_t = w_c/u_*$. which compares the rise speed to turbulence
intensity (c.f. Gopalan et al., 2008 who produced similar scaling for oil droplets but different to Daly
1984 who used a molecular diffusion equivalent). Large $Pe_t$ implies particle buoyancy dominates over
turbulence so that in the absence of growth, $Pe_t \gg 1$ indicates that settlement outstrips the redistribution
through turbulence. On the other hand, $Pe_t < 1$ describes where the crystals are continually being
redistributed throughout the water column regardless of their intrinsic rise speed (c.f. Rees Jones and
Wells 2018 frazil "explosion"). Dissipation rates in the range $\varepsilon = 10^{-7}$ to $10^{-6}$ W kg$^{-1}$ over a vertical scale
of 10 m results in an estimate of u*≈ 0.01-0.02 m s$^{-1}$ which is larger than both the average and
maximum observed rise rates and suggests a small apparent $Pe_t$. Placing the combined upper ocean
turbulence and crystal size data in this $Pe_t$ context (Figure 11), it appears that the average crystals and
turbulence conditions result in a situation where the rise speed should more than exceed turbulence
motions. However, considering the regime of smaller crystals in more turbulent conditions, then the ε
would be comparable. Notably, all of the continuum crystals observed by Frazer et al. (2020) fall to the
far left (small crystal diameter) of the $Pe_t$ domain.

The apparent bi-directional variability in vertical motion (Figure 7) implies wave-type motion.
However, the density structure gives no indication of stratification in the upper 30-50 m that would
support internal waves. It is possible that slow horizontal advection of convective processes moving
down and upwelling flows past the observation location might also be a factor influencing the measured
quantities.



### 4.3 Does sediment play a role in the mechanics?


While suspended sediment has been implicated in atmospheric ice formation (Kulkarni et al., 2014), it has not been identified as a nucleation point for marine ice crystal growth. Daly (1984) indicated that the thermodynamics precludes sediment playing a lead role in nucleation. However, there may be other pathways i.e. through aiding aggregation of ice crystals that *can* support nucleation. Previous observations in the region (Robinson et al., 2017) had not provided strong evidence that suspended sediment plays an important role in the SIPL in the region, so we did not have equipment specifically designed for such conditions. Sediment can potentially enter the water column through several pathways. First it might be held within the ice sheet underside as it leaves the coast at the grounding line, at which point it melts out and enters the basal boundary layer. Second, in some circumstances it might lift off from the seabed as part of an anchor ice process. Finally, sediment could be transported north by wind. This aeolian pathway deposits material on the sea ice surface at which point it would then have to find its way through the sea ice via brine channels. Atkins and Dunbar (2009) identified wind-blown transport in the region and leading to enhanced sediment concentrations in the sea ice. Here the wind-blown sediment is partly marine in origin and moved upwards through surface ablation/bottom accretion processes (Figure 12a).




The coincidence of a substantial sediment load suggests that, in this location at least, this might enhance crystal formation. However, a ready source of sediment is potentially a rare phenomenon at the continent-wide shelf edge suggesting that it is not a major controller of crystal production. It remains highly likely at grounding line zones where a confluence of ice shelf water outflow, glacially-driven sediment supply, tidal and subglacial resuspension and supercooling-induced re-freezing all likely combine. Furthermore, substantial platelet ice layers exist in many ice shelf influenced locations where there is no obvious sediment source (Langhorne et al., 2015; Hoppmann et al., 2020). The intriguing speculation that follows from this is – what role might sediment play in crystal nucleation near ice shelf grounding-lines and would we be able to tell if this influences marine ice formation (e.g. Fricker et al., 2001)?



### 4.4 What are the large-scale implications of flow-crystal interaction?

Local datasets such as the present experiment need to be placed into regional and continent-wide perspectives. There is a clear regional bias due to the majority of data coming from only a few field locations (Hoppmann et al., 2020). Debenham's comment about the region being the "strangest on the planet" suggests there may be some unique features that can't be generalised. While he wrote this in 1965 when Antarctica was still very under-surveyed it is not clear that there are many locations quite like the McMurdo Ice Shelf. At the regional Southern McMurdo Sound scale, Grima et al. (2016) and Brett et al. (2020) surveyed the area of the present study using remote sensing instruments that can detect platelet-influenced sea ice. Their surveys supported the conclusions drawn from the in-situ






sampling of Langhorne et al. (2015) that showed how platelet ice mirrored a likely outflow plume from the McMurdo/Ross ice shelf system's western limit. The Langhorne et al. (2015) spatial distribution of SIPL in McMurdo Sound shows a tongue of thicker SIPL extending 50 km northward from the dirty ice region where the present 2015 results were collected. More recent survey work of Brett et al. (2020) shows a striking similarity between SIPL and the location of the Dirty Ice. However, this also

corresponds to where one would expect basal cavity outflow so supply of ISW is available (Hughes et al., 2014). Following this water north, a number of studies have concluded that the ice shelf water from the Ross/McMurdo system persists for a hundred or more km northward (Hughes et al., 2014; Robinson et al., 2014).

Richter et al. (2020) state that ice shelf-sea ice-ocean connections remain the major outstanding

challenge in models operating at the continental and global scale. Injection of the range of processes described here into modelling approaches that typically resolve scales around 2 km will be a challenge. A starting point might be the role present large platelets play in the McMurdo Sound polynya formation (Dai et al., 2020). Polynya processes in particular become critical as they represent spatially-constrained phenomena driven by short high energy wind events but pre-conditioned by ice shelf water, that then

drive formation of new sea ice. It is the by-product of this sequence that generates high salinity shelf water that ultimately has a global thermohaline impact. Modelling of such processes thus may potentially need to better account for the nature of ice shelf water plumes.

The topic is clearly still in a discovery phase with many fundamental questions remaining unanswered. This work suggests research themes for understanding sea ice formation near ice shelves

should focus on the role of convection driven by SIPL crystal growth in modifying the turbulence in the upper water column and the feedbacks to the turbulence. In addition, the possible links between availability of nucleating material, crystal production and fate need to be examined, especially as to how this might support the formation of large, suspended ice crystals.

*Acknowledgements.* We would like to acknowledge Professor Pat Langhorne for her leadership on the wider topic. Tim Haskell is thanked for his development of the K131 infrastructure program. We are grateful to Inga Smith, Pat Wongpan, Cecilia Bitz, Greg Leonard and Andrew Pauling for their assistance and insight. Antarctica New Zealand provided field support. Funding was provided by the Royal Society of New Zealand Marsden Fund, NIWA Strategic Science Investment Funding, N.Z.

Antarctic Research Institute, the Deep South National Science Challenge, and the New Zealand Ministry of Business, Innovation and Employment through the Antarctic Science Platform (ANTA1801).

*Data availability:* data will be made available through an open repository.





*Author contribution:* CS conceived the experiment, conducted the majority of the sampling and analysis and led the writing. NR contributed to the experiment design and writing, GO'C produced the crystal data and BG collected the timeseries data, and developed the sounder sampling approach.

*Competing interests:* The authors have no conflicting interests.

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





Figures

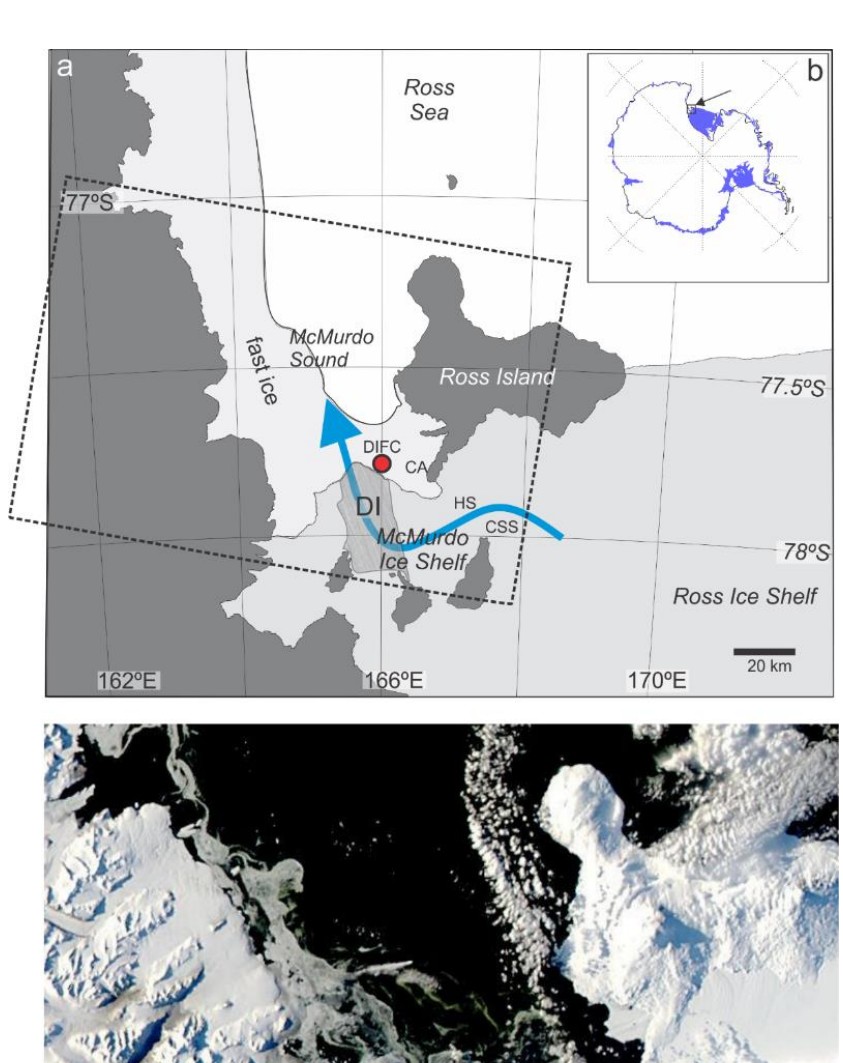

**Figure 1** Location details including (a) McMurdo Sound as an exit point for the Ross Ice Shelf
cavity (showing DIFC=Dirty Ice Field Camp, HS=Haskell Strait, CSS=Cape Spencer-Smith,
CA=Cape Armitage and the Dirty Ice=DI). The arrow shows the path of ice shelf water following
the complementary perspectives of Robinson et al. (2014) and Brett et al. (2020). (b) Antarctica
and ice shelf margin shown in blue with McMurdo Sound sub-region identified. (c) Modis
imagery from 2016 day 057 showing the dark smudge of the "dirty ice" (DI).





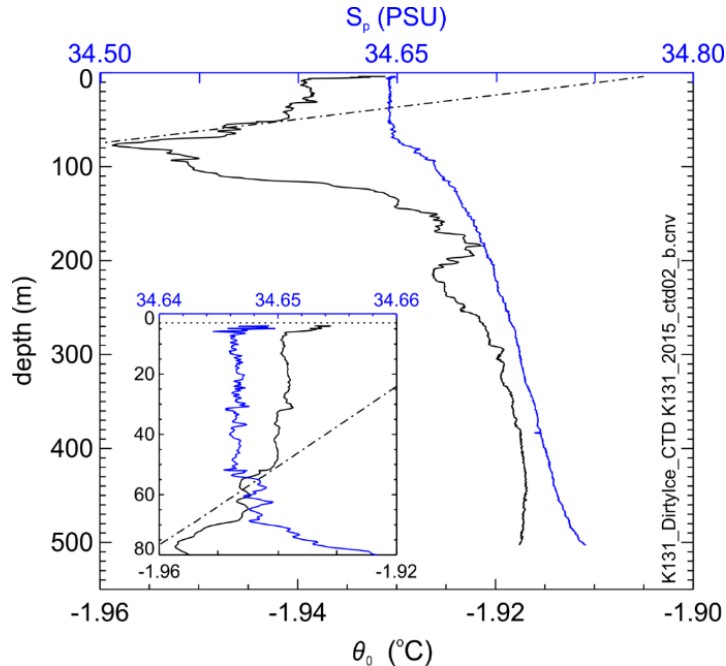

**Figure 2 Example vertical profiles of temperature and salinity from the CTD, along with the in situ freezing temperature. The inset shows an expanded view of the upper 80 m. This profile was from 0100 UTC on the 27th of November 2015.**

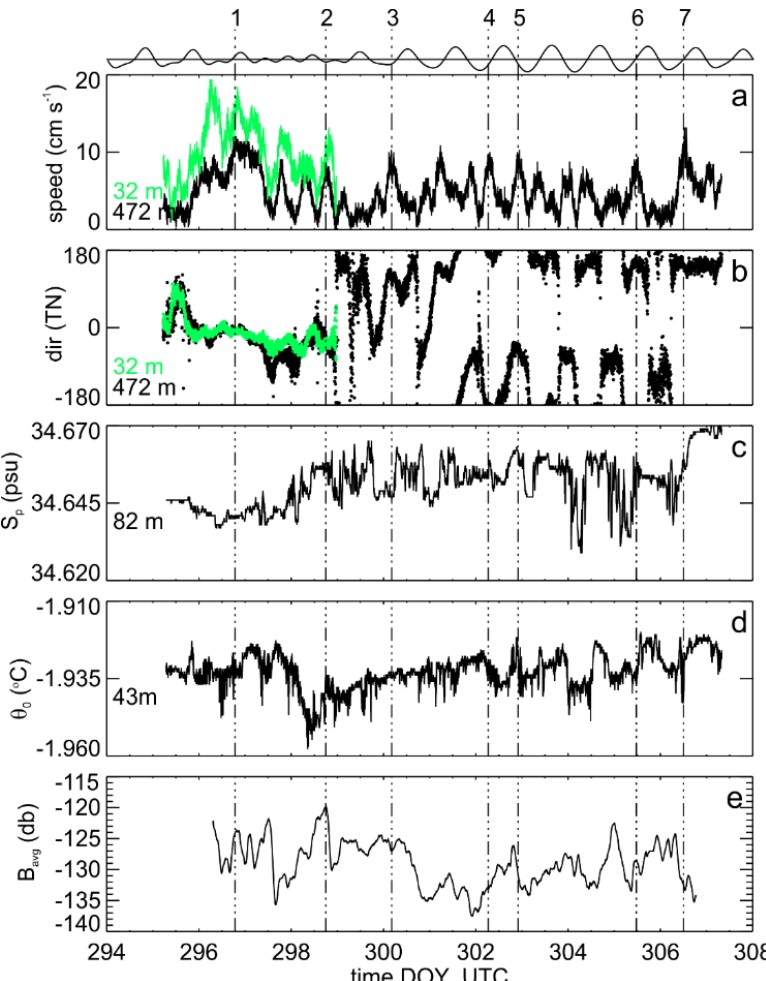

**Figure 3** Moored instrument data showing timeseries of (a) current meter speed at 472 m, (b) flow direction (towards) at 472 m, (c) salinity at 82 m, (d) potential temperature at 43 m and (e) vertically-averaged acoustic backscatter. Panels (a) and (b) include a short segment from the upper current meter (32 m) prior to ice-up. Selected peaks in speed are highlighted in all panels using vertical dashed lines and the modelled tidal height is traced above the top panel.




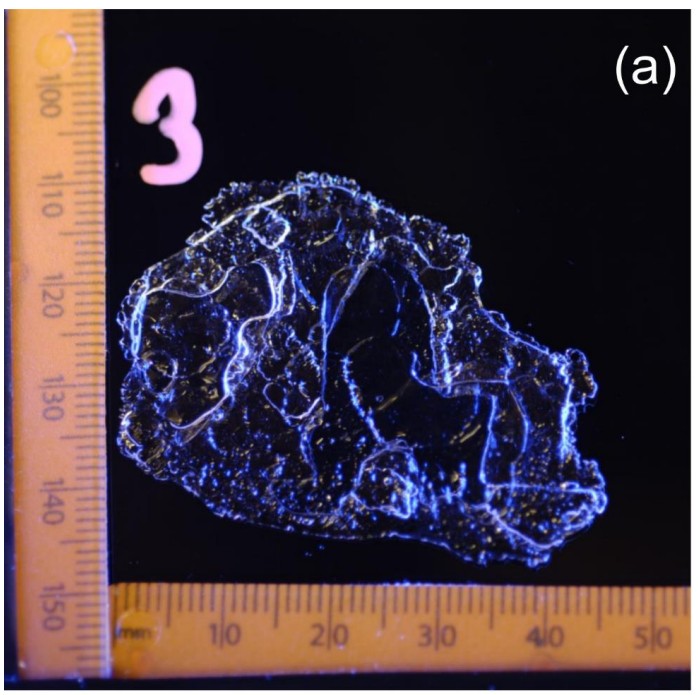

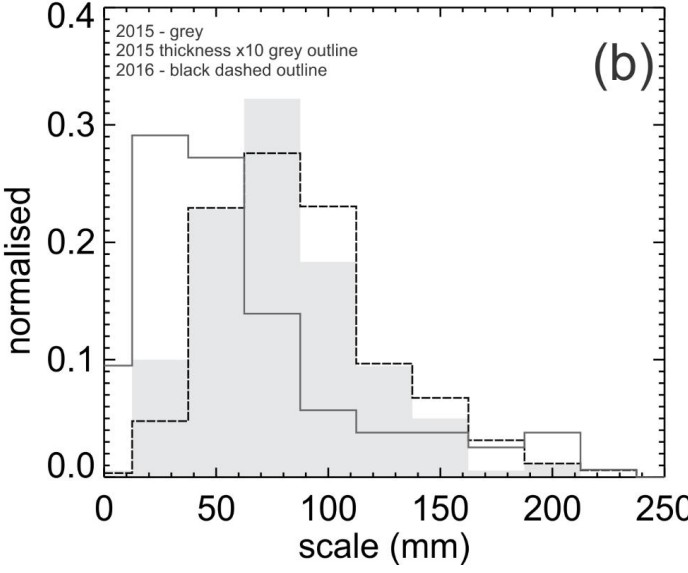

**Figure 4  Crystal details including (a) an image of an example platelet (ruler in mm).  (b) Scale distributions for 2015 (180 platelets) vector average of plan-view dimensions and the crystal thickness (scaled larger by a factor of 10) and 2016 (864 platelets, vector average of plan view dimensions).**




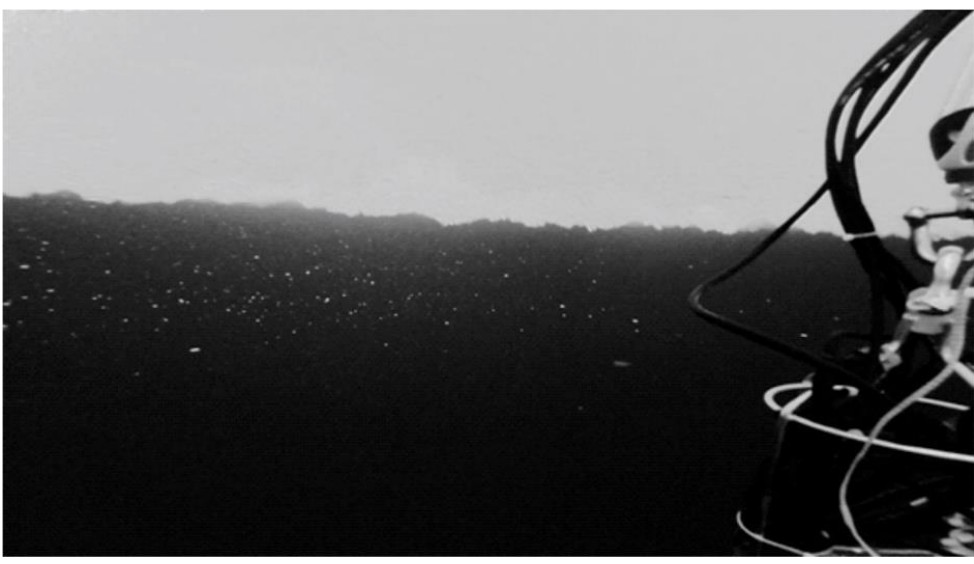

**Figure 5 video still beneath ice showing suspended crystals (approximate horizontal field of view 5 m). The image has had its contrast manipulated to aid in identification of the crystals.**





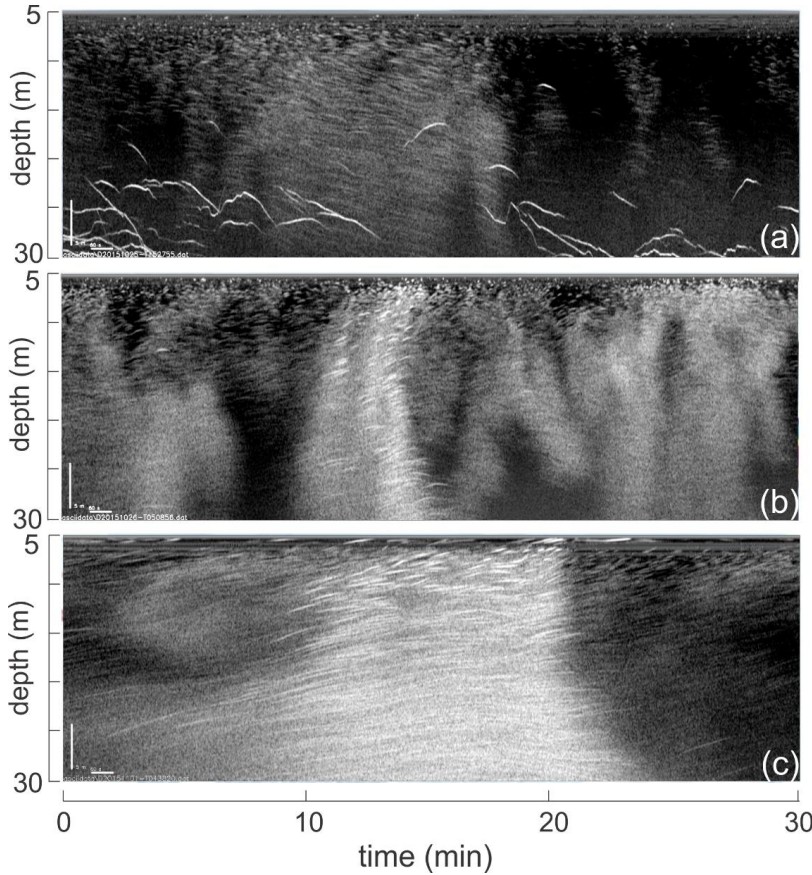

**Figure 6 acoustic sounder images from different times showing platelets rising from depth. Each panel is 30 minutes wide and 25 m deep (scale bar bottom left show 5 m vertical and 60 s horizontal scale). A diagonal corner-to-corner path represents a rise speed of 14 mm s⁻¹. The segments come from (a) 25oct2015 at 1528 UTC, (b) 26oct2015 UTC at 0509 and (c) 1nov2015 0438 UTC.**





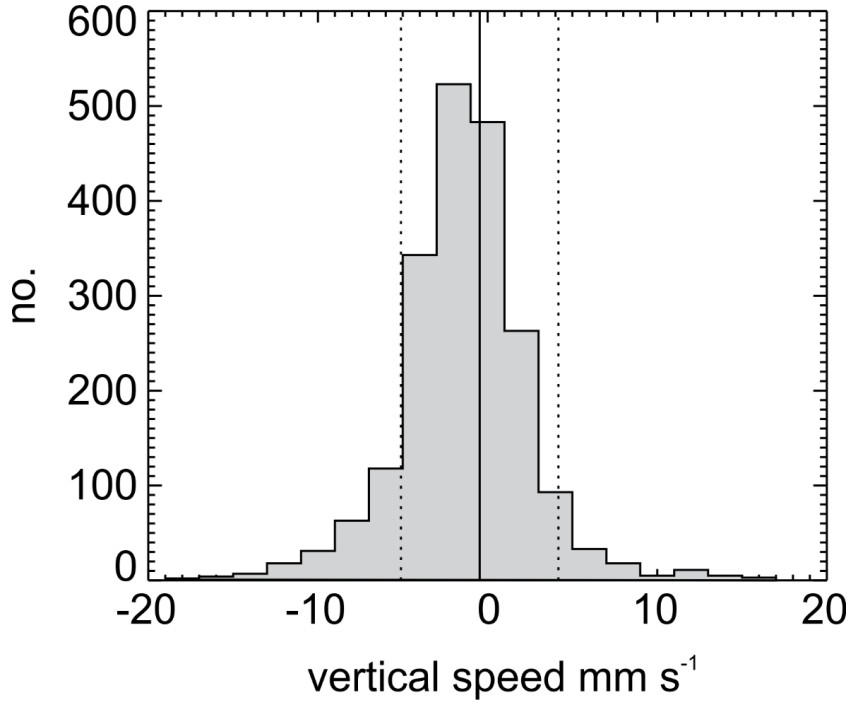


**Figure 7 vertical velocity derived from echograms, where the solid vertical line is the average value and the dashed lines are +/-1 one standard deviation. Negative speed is upwards.**




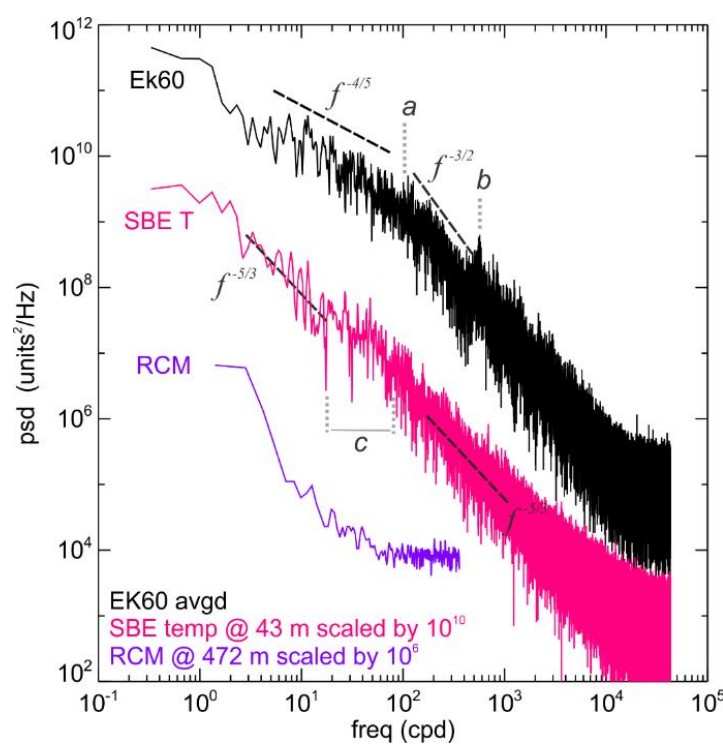

**Figure 8 Frequency distributions displayed using apparent power spectral densities showing the vertically averaged acoustic backscatter (EK60), the temperature at 43 m and the current speed at 472 m. The spectra are scaled to sit close by one another. Indicative frequency slopes and**
**annotation are discussed in the text.**






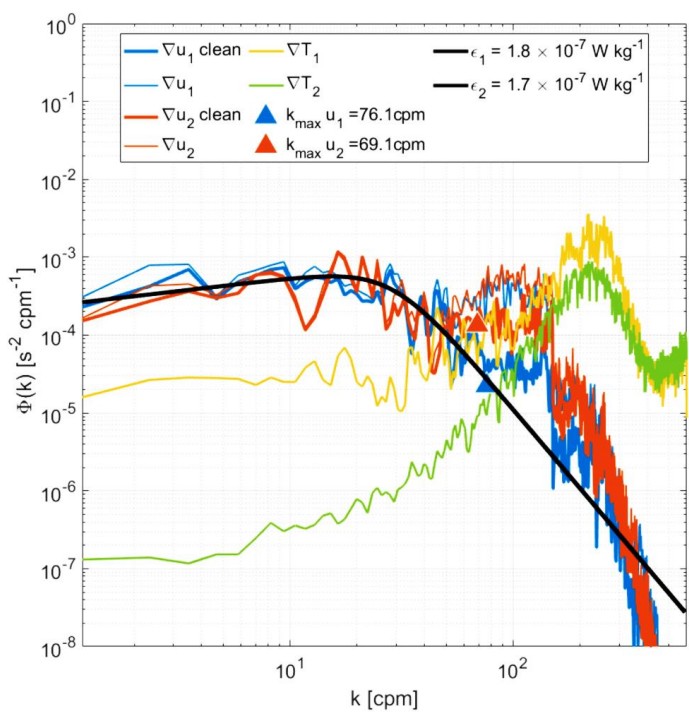

Figure 9 VMP microstructure example spectra showing temperature gradient ($\nabla T_{1,2}$) and microscale velocity shear ($\nabla u_{1,2}$) where the thin and thick lines show raw and filtered response. The black line is modelled energy spectra distribution for $\varepsilon$=1.6-1.7x10$^{-7}$ W kg$^{-1}$. The signal to the right of the triangles is amplified noise (Wolk et al., 2002).





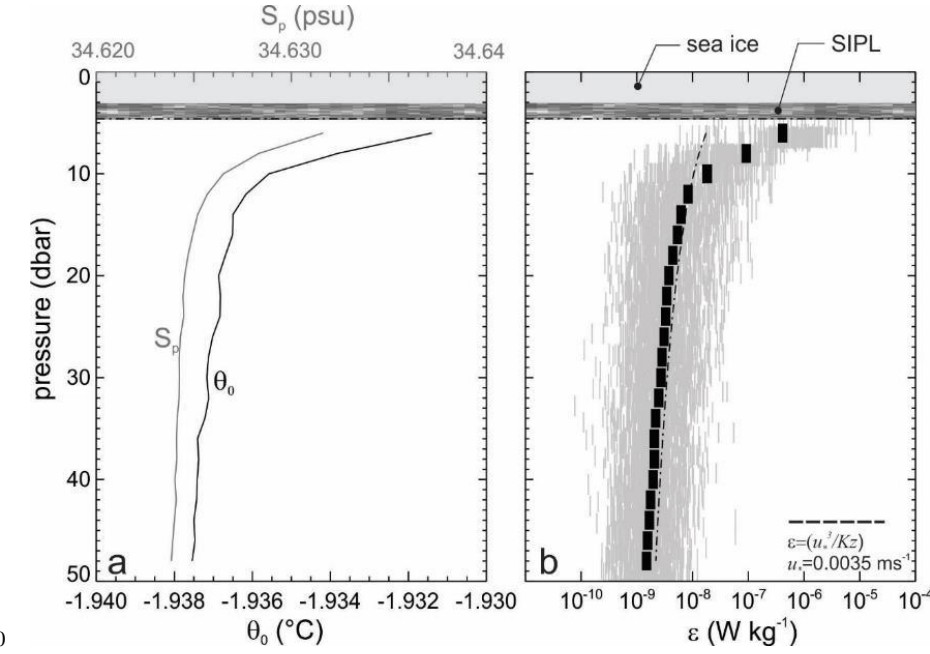


**Figure 10 Averaged microstructure profile data over the top 50 m showing (a) time-averaged potential temperature ($\theta_0$) and salinity ($S_p$) and (b) time-averaged turbulence dissipation rate, the individual samples and the shear-only scaling for $\varepsilon$. Both panels include the sea ice beneath hydrostatic zero and SIPL.**






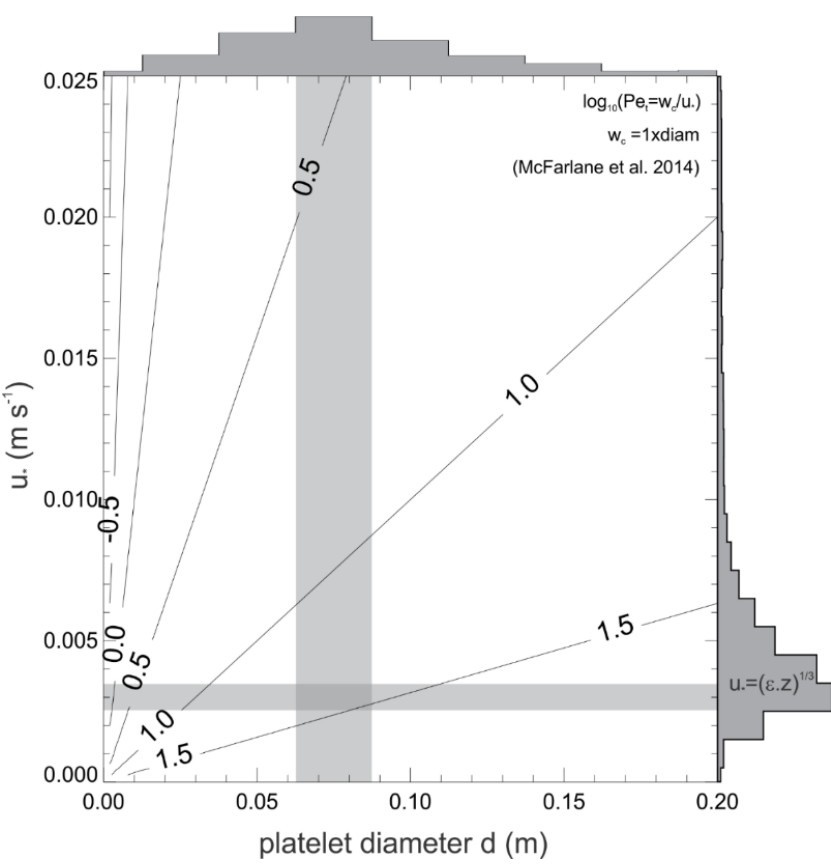

**Figure 11 Variation of $\log_{10}(Pe_t)$ as a function of platelet size (i.e. diameter) and boundary layer turbulent velocity scale u∗ showing approximate distribution and location of present results. The key assumption here is that the rise speed scales with diameter (see text) based on freshwater results from McFarlane et al. (2014). The diameter distribution is taken from Figure 4 and the u∗ distribution is from Figure 10 assuming the velocity scales with $(\varepsilon z)^{1/3}$.**




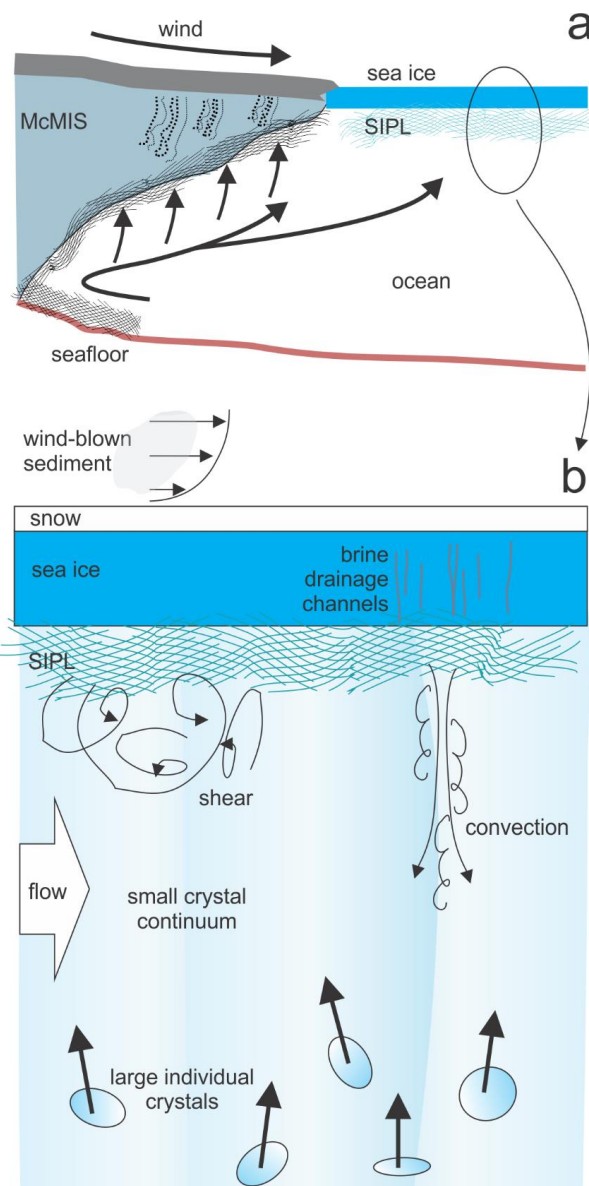

**Figure 12 (a) elevation sketch of the ISW cavity and plume and how it might interact with a sediment-cycling system. (b) Ice/SIPL/Ocean boundary layer structure.**