# Peer review of "Dynamics of Large Pelagic Ice Crystals in an Antarctic Ice Shelf Water Plume Flowing Beneath Land-Fast Sea Ice"

_The Cryosphere, 2020_

## Referee Comment (RC1) · Anonymous Referee #1 · 11 Nov 2020

Summary:

This manuscript summarizes measurements made under fast ice during a field campaign in McMurdo Sound. The authors find relatively large and thin ice crystals with an average diameter of about 100 mm directly below the main sea-ice layer. They provide visual and echo sounder evidence for particles further down in the water column and use the echo sounder data to derive vertical velocities of these particles. They argue that parts of these particles in the water column are the same ice crystals as observed directly under the sea ice. In addition, they report measurements of horizontal velocity and turbulence in the water column below the ice and attempt to relate the turbulence

measurements with the ice crystal size and vertical motion.

General comments:

This manuscript attempts to address 4 key questions around the growth, movement, and aggregation of large ice crystals or platelets in the water column as stated at the end of the introduction. The first two key questions related to the ice crystal size and their dependence on turbulence could in principal be addressed to some extent by the data and methods presented. However, the analysis lacks depth and conclusions seem largely unsupported. Much of the analysis and description are qualitative rather than quantitative and choices of depth levels, profiles, example images, ranges, etc. seem somewhat arbitrary. The results are structured in the order of the instruments that were deployed rather than addressing the scientific question. In many instances I am missing the context of a specific measurement and its overall use to support and address the key questions. The first key question is in my view not really novel, as large ice platelets have been observed in this region before. The second key question of how the crystals and their deposition depend on the vertical motion and turbulence would be novel, but the analysis appears not to be conclusive and provide new insights. The third and fourth key question on the source of crystal growth and its influence on the large scale are not even addressed by the analysis and only part of a somewhat inconclusive literature discussion. Overall, while being an interesting research topic, the analysis presented in this manuscript lacks scientific insight, support for its arguments, and a clear formulation of its conclusions.

The manuscript suffers from poor scientific writing with numerous incomplete thoughts and speculations. Rather than guiding the reader through the story line, results, and argumentation, the text lacks clarity and context. In many instances the context is implied rather than explicitly formulated, which makes it overall very difficult to read and understand the manuscript. It reads more like a report from the field campaign rather than a scientific paper.
I provide a number of specific comments illustrating the above mentioned issues below. But this list of comments is far from complete given the overall concerns that I have with the manuscript. In my view, the manuscript would need to be re-written much more carefully with the development of a scientific story line, a narrower focus on the actual subject and questions that can be addressed with the data available, a much deeper analysis, and a clear formulation of the conclusions that can be drawn from the analysis. At this point I recommend a rejection of the manuscript.

One possible recommendation to the authors would be to make better use of the echo sounder data to only address their question 2, which seems to be the only question that can be addressed with the presented data. They could provide a more in-depth assessment of uncertainties and more information of the apparent classification that has been performed, the particle size estimation from these data, and the attribution to ice crystals vs. other particles in the water column. There is a lot of unnecessary discussion that could be substantially reduced to strengthen the focus on this aspect of the paper and provide supporting evidence.

Specific comments:

1. The abstract completely misses any context, problem statement, or conclusions. It is rather a list of the measurements made with some results.

a. Line 11: What is "outflow" referring to? I assume Ice Shelf Water.

b. Line 13-14: "Advecting" horizontally or vertically? How can they be advected if they are already in a depositional layer? I assume this is not meant but the writing is ambiguous.

c. Line 17-18: There is no evidence provided in the manuscript that the flow is really of "tidal" nature. Is 0.1 m s-1 referring to vertical or horizontal velocity? The turbulence in the boundary layer can also result from other factors than tidal flow.

d. Line 19-21: How do class 1 and 2 differ? Both appear to be large particles. There is

no description in the manuscript of how these classes are being derived and how one of them is being attributed to ice crystals.

e. Line 21-23: Large ice crystals (platelets) have been observed before in this region. This is not a new finding. There is no evidence in the paper that these crystals are depositing as compared to growing locally. It is unclear what is meant by "fully grown". Is there a limit in the size to which the particles can grow? And how is this determined? There is no evidence in the paper for the evolution of growing particles and an upper limit.

f. Line 23: The histogram (Figure 4b) suggests a larger size than the 30-80 mm reported here in the abstract. The main text states an average of about 100 mm. All these numbers appear to be inconsistent.

g. Line 24: The "settlement" is not being clearly addressed in this paper. What are the implications? If there are any they should be explicitly stated here.

2. The introduction lacks clarity and focus. It is unclear what the gap of knowledge is, how it relates to the larger picture, and how it is being addressed.

a. Line 28-29: I am not sure how Antarctic sea ice variations are "confounding communication of key issues to stakeholders and decision makers" and what the formation of platelet ice has to do with it? This argument seems a bit arbitrary and poorly motivated.

b. Line 29-30: The warming on the deep water that is responsible for the increased melting of some of the ice shelves has not yet been attributed to "anthropogenic" sources. Thus, this statement is not quite correct. In addition, I do not know of any evidence that the Ross Ice shelf and the study site have been affected by this process.

c. Line 35: Not all ice shelf water rises buoyantly to the surface. In some regions the water flowing out of the cavity is actually denser than the surrounding water and sinks along the bottom (e.g. in Filchner Trough).

d. Line 47-48: The relation between ice growth, viscosity, and advection is unclear in

this sentence.

e. Line 53-56: It is unclear what is meant with this sentence.

f. Line 65-74: The purpose of this paragraph is unclear. I assume that it aims at providing context for ice-nucleation particles, but this is not clear to the reader at first. In addition, there seems to be irrelevant information and colloquial language. The reasoning implied by the last sentence is not clear to the reader. Isn't it well known by in-situ observations that there is supercooled water under the ice shelf? What is the relation between the supercooled water and the marine sediments at the surface of the ice shelf?

g. Question 1 is already known. There are large ice platelets in this region. Question 2 seems like the most appropriate for this study. Questions 3 and 4 are not addressed by this study.

3. The method of how the vertical velocity is being derived from the echo sounder data is not sufficiently described to be reproducible (lines 134-141). How large are the vertical segments? How are features identified and tracked? How well does the feature tracking from one time step to the next perform and what is the estimated error? The meaning of the sentences in lines 138-141 is unclear and needs clarification.

4. The importance and relevance of the background conditions described in section 3.1 for addressing the key questions is not evident from the text. How do the T and S profile, and current measurements affect where and how the ice crystals could grow in the water column? What is the role of different flow speeds at depth and close to the surface? Why are the depth levels chosen this way? How does the rising of the ISW plume cause a shear in the flow (line 175-176)? Doesn't the below surface freezing point temperature throughout the water column suggest that ISW is present in the entire water column? How else would it be possible to form waters with a temperature below -1.9degC down to 500m? It is unclear why tides should be the main driver of the horizontal flow (lines 176-181)? How about pressure/density gradients, wind forcing,

and large-scale circulation?

5. Section 3.2 contains many unsupported statements.

a. There is no apparent evidence for the reader and no scientific analysis provided in the manuscript that there is a "constant supply of crystals from depth" or their size (lines 188-189).

b. The authors do not provide evidence for the claims made in lines 190-195.

c. How is it known what fraction of the signal derived from the echo sounder results from ice crystals (line 196-197)? How is it known that the signal also includes biological particles? How is the background defined (lines198-201)? There is no evidence provided for rising particles (line 201-203). No analysis of video sequences supporting the identification of ice crystals (line 203) is provided.

d. Is the positive shift of the histogram towards an upward directed velocity statistically significant and larger than the uncertainty (lines 204-205)? If not, no such claim can be made.

e. How is the crystal size (7 cm, line 205) of the particles in the echo sounder estimated? Why is no histogram provided for those particles? What is the range of Reynolds numbers given the range in velocity and particle size?

f. What is the depth range over which the echo sounder data is evaluated (line 196ff)?

g. Lines 213-226: It is unclear how the discussion of the temporal evolution in these signals relates to the question addressed in the paper.

h. It is unclear how the conclusion is drawn that the "current meter spectrum is dominated by the tide" (line 228-229) and how this relates to the issue addressed in the paper. What is the role of the frequency spectrum for the ice crystal formation and transport and how is this analysis motivated?

6. Line 265: There is no evidence provided in the paper for the visual observations

referred to here.

7. Line 277-279: This sentence is not supported by the analysis. There is no evidence provided for the upward transport of large ice crystals and their origin.

8. Line 280-282: How is this separation obtained? Why is there no analysis showing the statistics of the different classes? What criteria are used to separate these classes.

9. Line 396: There is no evidence provided in the manuscript for a substantial sediment load in the region of ice crystal formation.

10. The manuscript is missing a clear formulation of the conclusions at the end.

Technical comments:

- "Pelagic": I am confused by the use of this word in the context of ice crystals. It is typically used for marine habitats and (if I am not mistaken) is a greek word for "open ocean". So, what is the intention of classifying ice crystals as "pelagic"? I thought the key aspect of the paper was to look at large size of the ice crystals occurring in this region.

- Line 48-49: "thought to exist" requires a reference.

- Line 78-80: These claims require references.

- Line 187: there is no figure 4c

- Line 409-410: A quote requires accurate referencing.

---

## Referee Comment (RC2) · Anonymous Referee #2 · 24 Nov 2020

Summary: In McMurdo Sound, supercooled ISW containing suspended frazil/platelet ice crystals circulates out from the McMurdo Ice Shelf cavity. The frazil/platelet ice crystals can then be deposited beneath the land-fast sea ice in McMurdo Sound and contribute to the sea ice cover and amass into an unconsolidated sub-ice platelet layer (SIPL). The objective of this study was to develop the understanding of ISW outflow and boundary-layer processes influencing the SIPL by combining coincident observations of ocean currents, Ice Shelf Water properties, turbulence structure and frazil/platelet ice suspended in the water column beneath land-fast sea ice in McMurdo Sound. Individual ice crystal dimensions were measured, and ice crystal behaviour monitored in the boundary layer beneath the SIPL with camera observations and acoustic backscatter.

[Figure]

Microstructure profiles were carried out to quantify turbulent mixing with the boundary layer. The observations were then related to background coincident ocean currents, water column properties and the tides. Three backscatter regimes identified were suggested to consist of 1) larger freely-floating platelet ice crystals, 2) intermediary size frazil to platelet ice crystals, and 3) a variable background backscatter assumed to be frazil ice suspension. Individual platelet ice crystals varied in size from 20-100 mm.

General Comments: ISW circulation processes and the deposition of frazil/platelet ice crystals at the interface between the ocean and SIPL are not well observed and thus not well understood. The processes influencing and forming the SIPL are thus very difficult to constrain. This paper presents important observations/analysis of coincident ocean properties, currents, boundary-layer processes and frazil/platelet ice beneath land-fast sea ice in a region of significant supercooled ISW outflow in McMurdo Sound. This is a rare dataset that provides much-needed information about the processes at play beneath the SIPL. However, I have several major comments that should be addressed before this paper is accepted to 'The Cryosphere'.

1) My main comment is that the research question is not clearly set up in the introduction and nor is it explicit in the abstract. This is a consistent thread throughout the paper and I had to read it several times before I could extract what this research aimed to address and why; why these specific measurements were made; and what their relevance was in the context of the research question and the greater motivation of the study.

The authors could address this by better framing and providing clarity on the motivations underpinning the research objectives. Additionally, spelling out why these specific measurements were made (especially for the non-expert in physical oceanography), what information they provide, and the significance and relevance of that information.

It would be useful to highlight the rarity of the observations and the importance of frazil/platelet ice size distributions/concentrations for understanding ISW plume dy-
namics and for informing relevant models such as described in Hughes et al. 2014 and Cheng et al. 2019.

2) I find the 'Dirty Ice' sedimentation hypothesis contradictory. In the introduction, the authors state that sediment in the ice is partly marine in origin, and likely entrained in marine ice as it formed on the underside of the ice shelf and that this is evidence of significant ISW outflow from this part of the McMurdo Ice Shelf. They then suggest that the sediment is providing seed material for frazil ice nucleation and could be a factor in the SIPL distribution beneath the land-fast sea ice in McMurdo Sound. My understanding is that previous oceanographic, sea ice core and SIPL measurements have shown that the SIPL distribution is driven by significant ISW outflow in this part of the sound.

More to the point, there are no observations or any analysis of suspended sediment load/particulates or frazil ice nucleation in this study and though an interesting discussion, section 4.3 seems superfluous and not directly relevant to this work. I recommend that this section is removed or supported with evidence.

3) My final comment is that the paper should include a conclusion section providing a clear summary of the background context and motivation for the study, what was done and why, the main results and the significance and relevance of the findings. I struggled to extract the main findings of this work.

Specific Comments: (L refers to Line numbers)

Title: I suggest changing 'large pelagic' to platelet ice crystals

1) L 10-11: Could you state why were these observations are presented and when they were collected?

2) L 23: I suggest changing 'fully grown" to larger platelet ice crystals of ? cm dimension.

3) L 23: What dimension are you referring to by 'Crystal scales'?

4) L 31-32: I suggest changing to 'a potential driver contributing to. . ..'.

5) L 33-34: This sentence needs more detailed information.

6) L 37: Sustained melting and/or decay of what? This sentence needs a reference.

7) L 41-42: I do not fully understand this statement. Can you develop?

8) L 46-56: I suggest including the work of Cheng et al. 2019 and emphasising the importance of suspended ice crystal load for ISW plume dynamics and for understanding the processes that influence the SIPL.

9) L 65: I suggest changing camps to observations.

10) L 67: Hunkeler et al. 2015 and Hoppman et al. 2020 are not specific to SIPL studies in McMurdo Sound.

11) L 75-85: This paragraph should be developed to set up the research question better and to describe what was done and why.

12) L 89: 'over an ocean depth'

13) L 103: Frazer et al. 2020 is in review.

14) L 107: I suggest providing the dimensions of the ice hole.

15) L 110-111: Does this statement refer to platelet ice and frazil ice crystals? This needs clarification.

16) L115: Is there any previous evidence that it is a tightly interlocked matrix?

17) L127-131: Where is the echosounder pointing? Upward to the SIPL base? Is the beam 3 m wide at 25 m depth or at the SIPL base? It would be very useful to have an estimation of the field of view.

18) L139: What is meant by superficial appearances? Can you develop and better clarify this statement?

19) L143-154: When were these measurements carried out and at what point on the tidal cycle?

20) L 165-172: Again, when were these measurements carried out and at what point on the tidal cycle?

21) L 175-176: What indications are there that this be the result of buoyantly forced ISW plume?

22) L 176-178: At what point on the tidal cycle were speeds of 0.1 m/s observed and in what direction? Please see comments provided below on Figure 3. It would be useful to provide date and month in parentheses after Julian Day.

23) L 180: I found the away and toward ice shelf convention confusing in the text and very difficult to interpret in Figure 3.

24) L 183: Is there a characteristic shape of platelet ice crystals?

25) L213-226: Can you be more specific about flow direction, point on the tidal cycle, temperature/salinity/backscatter changes and provide a date/time for each DOY.

26) L 230: What is the relevance of the frequency structure?

27) L 242: What is meant by 2-5 m undulation in SIPL underside? Where, when and how was this observed?

28) L 251: I suggest using profiles instead of realisations.

29) L 261: 4 Discussion

30) L293: I suggest rewording the last sentence of this paragraph.

31) L307-308: The Hoppmann et al. 2015 and Hunkeler et al. 2015 studies were not carried out in McMurdo Sound.

32) L 380-405 and L 417-420: As above in the specific comments about the co-occurrence of the 'Dirty Ice' and SIPL distribution.

33) L424: Richter et al. 2020 is in review.

34) L424-432: This paragraph could be developed further to provide a more substantiated context and broader outlook for this study.

35) L 427: I do not understand this statement.

Technical Comments: 1) The Frazer et al. 2020 and Richter et al. 2020 studies are currently in review.

Specific Comments on Figures

Figure 1: I suggest combining a and c into one figure and masking out the land. A MODIS optical image showing the land-fast sea ice conditions at the time of the field campaign would be more informative as would stating the date/month of acquisition.

Figure 2: At what point on the tidal cycle was this profile taken? And what are the DOY to relate to Figure 3?

Figure 3: I suggest moving a, b, c and d labels to top LH corner and put in parentheses; to make y-axis of tidal height larger. How was tidal height modelled? Please clarify convention for directions in a and b.

Figure 6: State DOY to relate to Figure 3.

Figure 12: Modify to account for changes to section 4.3 etc.

Suggested reference: Cheng, C., Jenkins, A., Holland, P. R., Wang, Z., Liu, C., and Xia, R.: Responses of sub-ice platelet layer thickening rate and frazil-ice concentration to variations in ice-shelf water supercooling in McMurdo Sound, Antarctica, The Cryosphere, 13, 265–280, https://doi.org/10.5194/tc-13-265-2019 , 2019.

---

## Author Comment (AC1) · 3 Dec 2020

**Response to Anonymous Referee 1

**Au:** We thank the Reviewer for the time and effort in commenting on our manuscript and pleased they found it an interesting topic.

We respectfully believe the Reviewer misunderstood the novelty of the dataset – the crystals that we focus on are free-floating and relative to "normal", very large. While the manuscript was clear on these points, we have made some additions to further emphasize this point.

[Figure]

The Reviewer doesn't like the clear separation of Results and Discussion we have chosen here. This is a style choice and we don't agree with the Reviewer's statement that it turns the manuscript into a field report. We are seeking to synthesize independent measurements to draw a conclusion based on multiple information channels. It is notable that the Reviewer makes no comment on the very difficult to obtain turbulence data that provide one of the key datasets and substantial novelty.

The Reviewer dismisses a number of points but provide no references to support their claims. In addition, in several places the Reviewer misrepresents the manuscript by criticising things we didn't say or saying we didn't do things and we clearly did.

The Review paradoxically suggests only a small component of the work being on-topic but then questions why there is no wider context. We attempt to respond to this below.

A number of points are of course helpful, and we respond to these as best we can below.

**Rev1:** General comments: This manuscript attempts to address 4 key questions around the growth, movement, and aggregation of large ice crystals or platelets in the water column as stated at the end of the introduction. The first two key questions related to the ice crystal size and their dependence on turbulence could in principal be addressed to some extent by the data and methods presented. However, the analysis lacks depth and conclusions seem largely unsupported. Much of the analysis and description are qualitative rather than quantitative and choices of depth levels, profiles, example images, ranges, etc. seem somewhat arbitrary.

**Au:** We are not sure why the Reviewer would make these statements – these are the first quantitative description of crystal sizes in the region we are aware of (there is nothing out there like Fig 4b we are aware of). We provide an extensive data record of turbulence beneath the sea ice (something rarely seen) and we provide a quantitative in situ estimate of crystal rise speed. We have added in some clarifying phrases around sampling choices.
**Rev1:** The results are structured in the order of the instruments that were deployed rather than addressing the scientific question. In many instances I am missing the context of a specific measurement and its overall use to support and address the key questions.

**Au:** Our present approach is common-place and, in this case, is required. We have added a clarifying sentence prior to the Questions in the Introduction.

**Rev1:** The first key question is in my view not really novel, as large ice platelets have been observed in this region before.

**Au:** The Reviewer is going to make such a sweeping comment they need to provide some references. They do not. Can they provide a reference that quantitatively identifies large free-floating crystals and measures aspects of their behaviour over any reasonable duration? Can they point to equivalent figures to Figs 4c, 5, 6, 7, 8 or 11?

It is clear that the Reviewer did not appreciate that the crystals are free floating. We used the term "pelagic" in the title and they question this later on. However, we use plenty of other ways of identifying this critical point of novelty including (i) 3rd line of abstract, (ii) a photograph, (iii) the second sentence of the Discussion, (iv) the very un-ambiguous Fig. 12… etc. It is worth noting that the recently published Frazer et al. GRL 2020) did not mention such crystals because they followed the false paradigm that these crystals don't exist in the water column.

Reviewer 2 seemed happy with the novelty and correctly suggests we need to reference Cheng et al 2019 (TC 2019). This recent modelling study notes a number of times they are limited by a paucity of data.

**Rev1:** The second key question of how the crystals and their deposition depend on the vertical motion and turbulence would be novel, but the analysis appears not to be conclusive and provide new insights.

**Au:** It is not clear on what basis the Reviewer makes these statements as they provide no references for where this work has been done previously – we measured the turbulence in the boundary layer and we measured the rise rate of the unusually large crystals and we measured the crystal sizes. Each of these processes has some variability which we quantified so we then put the combined measurements in a mechanistic framework and developed a non-dimensional approach to consider their behaviour. We provided conclusive comments about the range within which the behaviour falls.

**Rev1:** The third and fourth key question on the source of crystal growth and its influence on the large scale are not even addressed by the analysis and only part of a somewhat inconclusive literature discussion.

**Au:** We see that Reviewer Two didn't appreciate the sediment question and Discussion so we will remove this question along with some text and merge the rest. We do note that this concept is one the community is struggling with (Hoppmann et al 2020) and so circumstantial evidence would help guide the community to design the next sampling. We don't know what a "somewhat inconclusive literature discussion" means? We find it confusing to at once be criticised for not providing enough context but then also for examining how these data fit into a wider system context.

**Rev1:** Overall, while being an interesting research topic, the analysis presented in this manuscript lacks scientific insight, support for its arguments, and a clear formulation of its conclusions. The manuscript suffers from poor scientific writing with numerous incomplete thoughts and speculations. Rather than guiding the reader through the story line, results, and argumentation, the text lacks clarity and context. In many instances the context is implied rather than explicitly formulated, which makes it overall very difficult to read and understand the manuscript. It reads more like a report from the field campaign rather than a scientific paper.

**Au:** These are difficult criticisms to respond to given the Reviewer didn't understand the title or second sentence of the abstract or the first sentence of the Discussion or the very clear schematic diagram at the end or provide any references to back up their

claims.

We are not sure what the Reviewer means by a "story line"? We have some unique data and attempt to put them into a mechanistic and geographic context. It is unclear what the Reviewer means by "lacks clarity". We identify questions, we provide a description of the data collected in order to do our best to answer the questions. Then we do our best to connect the data to the questions and provide answers where we can.

**Rev1:** One possible recommendation to the authors would be to make better use of the echo sounder data to only address their question 2, which seems to be the only question that can be addressed with the presented data. They could provide a more in-depth assessment of uncertainties and more information of the apparent classification that has been performed, the particle size estimation from these data, and the attribution to ice crystals vs. other particles in the water column. There is a lot of unnecessary discussion that could be substantially reduced to strengthen the focus on this aspect of the paper and provide supporting evidence.

**Au:** We have added extra details on the echo sounder data relating to the details of the velocity estimation. We were surprised that the Reviewer made no comment on the turbulence data as being of significant value as the trajectories of the crystals are a balance of buoyancy and turbulence. There is little point exploring only one half of the balance to a greater depth.

**Rev1:** Specific comments: 1. The abstract completely misses any context, problem statement, or conclusions. It is rather a list of the measurements made with some results.

**Au:** The measurements are sufficiently novel we felt it was sensible to keep this as the focus and not provide some sweeping context. We have added opening sentences for clarity and incorporated responses to the below where constructive.

**Rev1:** a. Line 11: What is "outflow" referring to? I assume Ice Shelf Water. **Au:**

Amended

**Rev1:** b. Line 13-14: "Advecting" horizontally or vertically? How can they be advected if they are already in a depositional layer? I assume this is not meant but the writing is ambiguous.

**Au:** The sentence says "free-floating". We have modified the sentence in order to make this clear. It now says . . ." From a fast ice field camp, we captured the kinematics of free-floating relatively large (many 10s of mm in scale) ice crystals that were advecting and then settling upwards in a depositional layer on the sea ice underside (SIPL, sub-ice platelet layer)."

**Rev1:** c. Line 17-18: There is no evidence provided in the manuscript that the flow is really of "tidal" nature. Is 0.1 m s-1 referring to vertical or horizontal velocity? The turbulence in the boundary layer can also result from other factors than tidal flow.

**Au:** The abstract doesn't say the flows are of a tidal nature, it says what the tidal speed was. Furthermore, the Reviewer is incorrect that there is no evidence provided. Figure 3 shows the tides and related velocity structure. As well as non-tidal velocities, the manuscript described the impact of non-tidal turbulence including in Fig 12.

**Rev1:** d. Line 19-21: How do class 1 and 2 differ? Both appear to be large particles. There is no description in the manuscript of how these classes are being derived and how one of them is being attributed to ice crystals.

**Au:** We wouldn't describe motile biology as a particle. We added a clarifying sentence in the Discussion section 4.1 as it is beyond the scale of an abstract or the present focus.

**Rev1:** e. Line 21-23: Large ice crystals (platelets) have been observed before in this region. This is not a new finding. There is no evidence in the paper that these crystals are depositing as compared to growing locally. It is unclear what is meant by "fully grown". Is there a limit in the size to which the particles can grow? And how is this

determined? There is no evidence in the paper for the evolution of growing particles and an upper limit.

**Au:** The point is not that there are large crystals but that they are in suspension as clearly indicated. The Reviewer should include a reference if they are going to dismiss a significant amount of work. The evidence that they are depositing is because we observe them in the water and rising upwards towards a boundary where many more are to be found. The details the reviewer requests are beyond what would be expected in an abstract. We removed the phrase about "fully grown".

**Rev1:** f. Line 23: The histogram (Figure 4b) suggests a larger size than the 30-80 mm reported here in the abstract. The main text states an average of about 100 mm. All these numbers appear to be inconsistent.

**Au:** Fair point – we now consistently refer to the size range.

**Rev1:** g. Line 24: The "settlement" is not being clearly addressed in this paper. What are the implications? If there are any they should be explicitly stated here.

**Au:** We explicitly calculate the rise speed that drives settlement. The implications of having sea ice constructed from platelets are important and the focus of much study but not the focus here. We have modified the closing sentence of the Abstract. The Review is inconsistent as it asks for implications but criticises the inclusion of Question 4 around large-scale implications.

**Rev1:** 2. The introduction lacks clarity and focus. It is unclear what the gap of knowledge is, how it relates to the larger picture, and how it is being addressed.

**Au:** The concept of a "knowledge gap" is appealing but it's not really how geophysical science works. We don't know enough to say "we must go and measure process X". We can model the planet now – it's just that is doesn't match what we measure. It is instead a refinement of scales and mechanics sufficient to get the interplaying processes right.

The Introduction opens with the global setting and the first paragraph ends with "This

supercool water drives sea ice growth by absorbing heat into the stratified upper ocean and facilitates the generation and growth of ice crystals". The next paragraph opens with unknowns about the crystals. This ultimately leads to four questions one of which is what the knowledge gap is and another is essentially "how it relates to the larger picture" which we don't presage before we actually do the science. They have a balance moving from discovery through to implications. This is quite focused. We have re-worked aspects of the Introduction to see if we can meet the Reviewer's expectations.

**Rev1:** a. Line 28-29: I am not sure how Antarctic sea ice variations are "confounding communication of key issues to stakeholders and decision makers" and what the formation of platelet ice has to do with it? This argument seems a bit arbitrary and poorly motivated.

**Au:** Stakeholders want certainty and a sense we can model the earth system accurately. We, as a community, can't do this at present. However, expanding on this won't help the focus. We have added a linkage phrase and a reference to help clarify this important point for the Reviewer.

**Rev1:** b. Line 29-30: The warming on the deep water that is responsible for the increased melting of some of the ice shelves has not yet been attributed to "anthropogenic" sources. Thus, this statement is not quite correct. In addition, I do not know of any evidence that the Ross Ice shelf and the study site have been affected by this process.

**Au:** The text said neither of these things.

**Rev1:** c. Line 35: Not all ice shelf water rises buoyantly to the surface. In some regions the water flowing out of the cavity is actually denser than the surrounding water and sinks along the bottom (e.g. in Filchner Trough).

**Au:** A bottom-following ice shelf water plume must be unusual. A reference would

help.

**Rev1:** d. Line 47-48: The relation between ice growth, viscosity, and advection is unclear in this sentence.

**Au:** We have added a clarifying phrase. "If these crystals grow slowly, remaining sufficiently small that viscosity dominates buoyancy so that rise rates are very slow, then they are mainly passively advected."

**Rev1:** e. Line 53-56: It is unclear what is meant with this sentence.

**Au:** We have clarified this sentence. "Despite the challenges in making measurements in this environment, one correlation that emerges is that SIPL thickness and supercooled seawater are co-located (Langhorne et al., 2015; Brett et al., 2020)."

**Rev1:** f. Line 65-74: The purpose of this paragraph is unclear. I assume that it aims at providing context for ice-nucleation particles, but this is not clear to the reader at first. In addition, there seems to be irrelevant information and colloquial language. The reasoning implied by the last sentence is not clear to the reader. Isn't it well known by in-situ observations that there is supercooled water under the ice shelf? What is the relation between the supercooled water and the marine sediments at the surface of the ice shelf?

**Au:** The paragraph is to provide part of the "story line" asked for elsewhere. It provides the context for the geographical location and the datasets that allowed us to arrive at the point that we could collect the observations that we did.

Which language is colloquial? The text in quotes come from the literature as referenced. We have added some text to aid in understanding around the relationship between the supercooled water and the marine sediments at the surface of the ice shelf.

**Rev1:** g. Question 1 is already known. There are large ice platelets in this region. Question 2 seems like the most appropriate for this study. Questions 3 and 4 are not

addressed by this study.

**Au:**  As above, the Reviewer does not provide a useful reference for evidence of large free-floating crystals and we do not believe they have been adequately described previously. The Review includes multiple requests for context such as that provided by the Discussion of Q4.

**Rev1:**  3.  The method of how the vertical velocity is being derived from the echo sounder data is not sufficiently described to be reproducible (lines 134-141). How large are the vertical segments? How are features identified and tracked? How well does the feature tracking from one time step to the next perform and what is the estimated error? The meaning of the sentences in lines 138-141 is unclear and needs clarification.

**Au:**  These are useful clarifications. We have added some extra text and a reference here on the analysis. One of the neat aspects is the improvement in vertical resolution through vertical low-pass filtering. It is notable that the Reviewer focuses on this half of the mechanics and misses the important contribution from the turbulence data and the intriguing results around how the turbulence distribution doesn't perfectly match a friction driven boundary-layer.

**Rev1:**  4.  The importance and relevance of the background conditions described in section 3.1 for addressing the key questions is not evident from the text. How do the T and S profile, and current measurements affect where and how the ice crystals could grow in the water column?  What is the role of different flow speeds at depth and close to the surface?  Why are the depth levels chosen this way? How does the rising of the ISW plume cause a shear in the flow (line 175-176)?  Doesn't the below surface freezing point temperature throughout the water column suggest that ISW is present in the entire water column? How else would it be possible to form waters with a temperature below -1.9degC down to 500m?

**Au:**  These are interesting questions but not all relate to the focus at hand. The thick ISW layer is well known for this region (Robinson et al 2014) but with crystals forming
beneath the surface we are interested in insitu supercooling. We provide some extra information and clarify the point about the ISW plume in terms of what we actually said.

**Rev1:** It is unclear why tides should be the main driver of the horizontal flow (lines 176-181)? How about pressure/density gradients, wind forcing, and large-scale circulation?

**Au:** It is not clear the Reviewer read the text. It said "The deepest current meter provided the best quality current speed results (Figure 3a,b). The upper current meter did work for a few days at the beginning and was sufficient to show that the upper 175 speeds were between 50 and 100

**Rev1:** 5. Section 3.2 contains many unsupported statements. a. There is no apparent evidence for the reader and no scientific analysis provided in the manuscript that there is a "constant supply of crystals from depth" or their size (lines 188-189).

**Au:** This is why we provide the Results-Discussion structure as we do. We have 10 days of day sampled constantly with modest signal fluctuations. The text now references Fig 3e and we removed the phrase about scale.

**Rev1:** b. The authors do not provide evidence for the claims made in lines 190-195.

**Au:** We provide a Figure and a statement about what we are seeing in the figure and return to this in the Discussion. We could add arrows in the figure or mention a sequence of specific times? The Reviewer needs to provide more detail about what they would like to see in terms of evidence in order for us to extend this point.

**Rev1:** c. How is it known what fraction of the signal derived from the echo sounder results from ice crystals (line 196-197)?

**Au:** We don't claim to be able to quantify this. It remains a good question though. It doesn't impact our analysis because it responds to the peaks in backscatter anyway.

**Rev1:** How is it known that the signal also includes biological particles?

**Au:** We suggest they are biological in nature because they look like they're swimming

erratically and across the wider motion. The text now explicitly states this.

**Rev1:** How is the background defined (lines198-201)?

**Au:** This is essentially the same point as above where we noted the present analysis does not need to explicitly identify this separation to calculate the vertical motion. Instead the analysis is primarily responsive to the larger signals in the segments.

**Rev1:** There is no evidence provided for rising particles (line 201-203).

**Au:** Figure 7 shows an average rise speed. We can include video of crystals rising in the supplementary information if that would help.

**Rev1:** No analysis of video sequences supporting the identification of ice crystals (line 203) is provided.

**Au:** Correct -this is why we analysed the acoustic information which has a much better quantitative basis.

**Rev1:** d. Is the positive shift of the histogram towards an upward directed velocity statistically significant and larger than the uncertainty (lines 204-205)? If not, no such claim can be made.

**Au:** We have clarified the text on the reliability of the measurements which is aided by (i) pre low-pass filtering and (ii) averaging over the 10 days of sampling.

**Rev1:** e. How is the crystal size (7 cm, line 205) of the particles in the echo sounder estimated? Why is no histogram provided for those particles? What is the range of Reynolds numbers given the range in velocity and particle size?

**Au:** We re-wrote the Reynolds number paragraph to clarify these points. It now says "Considering the Reynolds number (Re=characteristic velocity x dimension/kinematic viscosity) as quantifying the balance of inertia and viscosity, and with the larger crystals being on average around 10 cm in diameter and rise speeds of the order of 1 cm s-1, this implies a Reynolds number ranging from 50-2000 with an average of 1000."

[Figure]

**Rev1:** f. What is the depth range over which the echo sounder data is evaluated (line 196ff)?

**Au:** The depth range shown in the Figure is representative of the usable data. The methods now explicitly states this depth in Section 2.3.

**Rev1:** g. Lines 213-226: It is unclear how the discussion of the temporal evolution in these signals relates to the question addressed in the paper.

**Au:** The temporal evolution relates to the consistency of supply which relates to the questions about presence, dynamics and wider implications.

**Rev1:** h. It is unclear how the conclusion is drawn that the "current meter spectrum is dominated by the tide" (line 228-229) and how this relates to the issue addressed in the paper. What is the role of the frequency spectrum for the ice crystal formation and transport and how is this analysis motivated?

**Au:** The Reviewer makes a fair point about the tides and spectrum. While the statement in the text was true, the displayed spectrum was derived in a way that provided best reliability in the upper frequencies as stated in the text and consequently did not resolve the tidal peak. This statement has been modified. The spectrum is there to see if there are any dominant modes of variation or if it is a continuum of scales in the driving velocity signal – and it is largely the latter. The text now says . . ."The current meter spectrum is constrained to lower frequencies with much of the spectrum above 50 cpd reaching an apparent noise-floor implying that the variations seen in temperature and backscatter are not advection-driven."

**Rev1:** 6. Line 265: There is no evidence provided in the paper for the visual observations referred to here.

**Au:** Figure 5 provides this evidence. We now explicitly reference Fig 5.

**Rev1:** 7. Line 277-279: This sentence is not supported by the analysis. There is no evidence provided for the upward transport of large ice crystals and their origin.

**Au:** The unique analysis in Figure 7 is that evidence.

**Rev1:** 8. Line 280-282: How is this separation obtained? Why is there no analysis showing the statistics of the different classes? What criteria are used to separate these classes.

**Au:** We have clarified the language here and in the Results. The separation of Class 1 is behavioural and there is probably a continuum between classes 2 and 3 but our acoustic sampling is dominated by the larger end of this combined class. However, the statistics of the different classes is not the focus here. Class 1 are biological in nature and Class 3 is well covered by Frazer et al. 2020. It is Class 2 that is not thought to exist in suspension that is the contribution here. The focus of the manuscript as identified in the Questions is around the large crystal behaviour in the turbulence of the ice shelf plume.

**Rev1:** 9. Line 396: There is no evidence provided in the manuscript for a substantial sediment load in the region of ice crystal formation.

**Au:** We have moved or deleted the sediment text in keeping with both Reviewer's comments. We do note however, that the field site is down-wind of the geographic feature called the "dirty ice" and we did provide circumstantial evidence – ". A hot-water cutter was used to melt through and remove the sea ice in blocks. It was notable that upon removal of the blocks, the water which filled the hole appeared milky but that this gradually dissipated over the subsequent days. After 12 days of operations and many seal occupations of the holes, the hole water was fully flushed and very clear. We speculate that the water in the hole was initially from the melting of the sea ice and upwards drainage from the SIPL and contained sufficient levels of sediment to be visible but that over time this was replaced with clear ocean water".

**Rev1:** 10. The manuscript is missing a clear formulation of the conclusions at the end.

**Au:** We thought having the closing thoughts as part of the large-scale context question was sufficient. However, motivated by the comments from the Reviewer we have added a "concluding remarks" subheading.

**Rev1:** Technical comments: - "Pelagic": I am confused by the use of this word in the context of ice crystals. It is typically used for marine habitats and (if I am not mistaken) is a greek word for "open ocean". So, what is the intention of classifying ice crystals as "pelagic"? I thought the key aspect of the paper was to look at large size of the ice crystals occurring in this region. –

**Au:** We used "pelagic" in the sense of away from boundaries. In order to avoid confusion we can retitle using the term "suspended" but even this has its issues as it implies that there is no net rise which is counter to our analysis. We could put "pelagic" in quotes?

**Rev1:** Line 48-49: "thought to exist" requires a reference. –

**Au:** This was all text associated with the recent review by Hoppmann and the Frazer paper both referenced at the start of the paragraph. We now repeat the references.

**Rev1:** Line 78-80: These claims require references. –

**Au:** This was all text associated with the recent review by Hoppmann referenced at the start of the paragraph and nicely synthesizing a number of studies and providing a nice context for this present work. We now repeat the references.

**Rev1:** Line 187: there is no figure 4c –

**Au:** Corrected thanks

**Rev1:** Line 409-410: A quote requires accurate referencing.

**Au:** We were returning to a point made earlier which was accurately referenced. We have repeated the reference.

---

## Author Comment (AC2) · 4 Jan 2021

**Author:** We thank the Reviewer for the time and effort in commenting on our manuscript and especially for their constructive suggestions and we respond to these as best we can below. We (i) modified the title, (ii) improved the clarity of the Abstract and Introduction, (iii) removed a theme on the role of sediment and (iv) clarified when various data were collected. In addition, we modified several of the Figures in response to their points. We believe this has resulted in significant improvements in the manuscript and the "rare dataset" it presents.

**Rev2:** General Comments: ISW circulation processes and the deposition of frazil/platelet ice crystals at the interface between the ocean and SIPL are not well observed and thus not well understood. The processes influencing and forming the SIPL are thus very difficult to constrain. This paper presents important observations/analysis of coincident ocean properties, currents, boundary-layer processes and frazil/platelet ice beneath land-fast sea ice in a region of significant supercooled ISW outflow in Mc-Murdo Sound. This is a rare dataset that provides much-needed information about the processes at play beneath the SIPL. However, I have several major comments that should be addressed before this paper is accepted to 'The Cryosphere'.

**Au:** We are pleased they found it "a rare dataset that provides much-needed information about the processes at play beneath the SIPL". There remains much to be discovered in the under-sampled Antarctic oceanic environment. It is noteworthy that the recent paper in TC by Cheng et al 2019 that the Reviewer brought to our attention closes with a call for more observations of exactly the kind we present.

**Rev2:** 1) My main comment is that the research question is not clearly set up in the introduction and nor is it explicit in the abstract. This is a consistent thread throughout the paper and I had to read it several times before I could extract what this research aimed to address and why; why these specific measurements were made; and what their relevance was in the context of the research question and the greater motivation of the study. The authors could address this by better framing and providing clarity on the motivations underpinning the research objectives. Additionally, spelling out why these specific measurements were made (especially for the non-expert in physical oceanography), what information they provide, and the significance and relevance of that information.

**Au:** Reviewer 1 made a similar comment and we have made efforts to improve this aspect. It is challenging for small-scale process studies to connect to the global system while retaining focus on the smaller scales. We are fortunate that the major review
by Hoppmann et al has just come out providing excellent context. We have added some motivational text right at the start. It now says "Here we examine the dynamics of large floating crystals observed to be suspended in the boundary-layer beneath a fast ice layer near a large Antarctic ice shelf cavity. This is unusual as one would expect the crystals to either be much smaller or rapidly float to merge into the underice aggregation of ice crystals. The existence and persistence of such large crystals must influence regional variability in Antarctic sea ice."

**Rev2:** It would be useful to highlight the rarity of the observations and the importance of frazil/platelet ice size distributions/concentrations for understanding ISW plume dynamics and for informing relevant models such as described in Hughes et al. 2014 and Cheng et al. 2019.

**Au:** This is a good point and we have added the following... "Is there evidence of large suspended crystals? While they are known to exist on the ice underside there is little evidence confirming their presence in the water column itself. If they can be identified this would be a novel contribution and aid future model development."

**Rev2:** 2) I find the 'Dirty Ice' sedimentation hypothesis contradictory. In the introduction, the authors state that sediment in the ice is partly marine in origin, and likely entrained in marine ice as it formed on the underside of the ice shelf and that this is evidence of significant ISW outflow from this part of the McMurdo Ice Shelf. They then suggest that the sediment is providing seed material for frazil ice nucleation and could be a factor in the SIPL distribution beneath the land-fast sea ice in McMurdo Sound. My understanding is that previous oceanographic, sea ice core and SIPL measurements have shown that the SIPL distribution is driven by significant ISW outflow in this part of the sound. More to the point, there are no observations or any analysis of suspended sediment load/particulates or frazil ice nucleation in this study and though an interesting discussion, section 4.3 seems superfluous and not directly relevant to this work. I recommend that this section is removed or supported with evidence.

TCD
**Au:** The point is that the situation needs both sediment as a nucleator AND the ISW to drive the freezing. However, in keeping with both Reviewers advice we have removed the sediment theme, modified the closing schematic and deleted or shifted the associated text.

**Rev2:** 3) My final comment is that the paper should include a conclusion section providing a clear summary of the background context and motivation for the study, what was done and why, the main results and the significance and relevance of the findings. I struggled to extract the main findings of this work.

**Au:** We had incorporated closure into the final paragraph but now have inserted a subheading title and more clearly reiterated the main points.

**Specific Comments: (L refers to Line numbers)**

**Rev2:** Title: I suggest changing 'large pelagic' to platelet ice crystals

**Au:** Rev 1 was confused by the pelagic term so we will replace this. However, as noted by the recent Hoppmann review "platelet" itself generates confusion so we will use "large suspended crystals". The revised title now says... "Dynamics of Large Suspended Ice Crystals in an Antarctic Ice Shelf Water Plume Flowing Beneath Land-Fast Sea Ice".

**Rev2:** L 10-11: Could you state why were these observations are presented and when they were collected?

**Au:** The text now says... "The connections between ice shelf cavities and sea ice influence sea ice development and persistence. One unique feature is the potential for sea ice growth due to crystal accretion. Here we present unique observations of boundary-layer processes and ice crystal behaviour in an Ice Shelf Water outflow region from the Ross/McMurdo Ice Shelves. From a fast ice field camp during the Spring
of 2015, we captured the kinematics of free-floating relatively large (many 10s of mm in scale) ice crystals that were advecting and then settling upwards in a depositional layer on the sea ice underside (SIPL, sub-ice platelet layer)."

**Rev2:** L 23: I suggest changing 'fully grown" to larger platelet ice crystals of ? cm dimension.

**Au:** The text has been modified to say... "This second class of backscatter was associated with crystal sizes far larger than typical, certainly larger than anything normally described as frazil. Measurement indicated crystal width scales of the range 5-200 mm with an average of 93-101 mm depending on the year."

Rev2: L 23: What dimension are you referring to by 'Crystal scales'?

**Au:** Crystal "width" as per above text. Note we enhance this aspect later on with an improved figure that now shows the distribution of the major and minor crystal dimensions.

Rev2: L 31-32: I suggest changing to 'a potential driver contributing to: : :.'.

Au: OK change made

Rev2: L 33-34: This sentence needs more detailed information.

**Au:** We were trying to constrain the Introduction to the boundary-layer processes at hand rather than a wide range of polar oceanographic processes. *"The meltwater exiting the cavity is typically very cold as it comprises water formed within the cavity at depth so that the local freezing point temperature is depressed due to the pressure. This water mixes with the ambient ocean resulting in a fresher, cold seawater plume that seeks out the fastest upward flow path on the shelf underside subject to the Coriolis*

TCD
force and basal slope (MacAyeal 1985; Jenkins and Bombusch 1995; Smedsrud and Jenkins 2005; Stevens et al., 2020)."

**Rev2:** L 37: Sustained melting and/or decay of what? This sentence needs a reference.

**Au:** The sentence has been reworked and references added. It now says "These plumes will develop depending on the balance of inflow, re-freezing either of the sea ice underside or in the formation of suspended crystals removing their thermal deficit, and mixing with warmer waters."

Rev2: L 41-42: I do not fully understand this statement. Can you develop?

**Au:** We merged it with the bracketing sentences so the section now says... "At this point the basal slope driver of flow ceases and the persistence of the supercool plume is controlled by initial buoyancy, growth of new ice, topography and mixing (Hughes et al., 2014) which are all changing relatively rapidly. This plume of supercool water influences sea ice growth by both enhancing upper ocean stratification and absorbing heat (Robinson et al., 2014; McPhee et al., 2016). Whether or not there is an associated phase change is critical and so the presence of ice crystals within the upper water column has a number of implications from thermal budgets, convection and ecological habitat (Hoppmann et al., 2020)."

**Rev2:** L 46-56: I suggest including the work of Cheng et al. 2019 and emphasising the importance of suspended ice crystal load for ISW plume dynamics and for understanding the processes that influence the SIPL.

**Au:** This is a good point thanks – we were not aware of the reference and had included a number of other older references. We now include this reference and note the point that it advocates for more observations of the processes described here.
Rev2: L 65: I suggest changing camps to observations.

**Au:** OK but observations doesn't quite serve the purpose as it is meant to identify a sequence of "experiments". We have amended to... "long sequence of sea ice camps in the McMurdo Sound region (Robinson et al., 2020) enabled observations that have revealed..."

**Rev2:** L 67: Hunkeler et al. 2015 and Hoppman et al. 2020 are not specific to SIPL studies in McMurdo Sound.

**Au:** OK - we were seeking to generalise but can split this to now have sentences on the local situation and also the wider Antarctic evidence.

**Rev2:** L 75-85: This paragraph should be developed to set up the research question better and to describe what was done and why.

**Au:** We have reworked this paragraph and the end of the previous paragraph to meet the Reviewer's point. The text now says... "Here we examine the dynamics of large ice crystals observed freely floating in the upper ocean and the turbulence within which they move in the context of background hydrographic conditions. Hoppmann et al. (2020) review our present understanding of Antarctic platelet ice and makes it clear the topic is still in a discovery phase – partly due to the challenges of making comprehensive observations. At the same time, modelling approaches have needed to advance – creating a tension in that not all the relevant processes and scales are known. Geophysical boundary-layers are well understood and it is known that ice crystals on the sea ice underside influences interfacial momentum transfer, sea ice composition and strength as well as ecological habitat throughout localised parts of Antarctic coastal waters. However, if these crystals can accumulate initially as large crystals rather than primarily grown once settled will make a difference to these processes. Consequently, the presence of large suspended crystals result in a number of questions that provide focal points for the present study. (1) Is there evidence of large suspended crystals?

TCD
While they are known to exist on the ice underside there is little prior published evidence confirming their presence in the water column itself. This observation is novel and provides a guide for future model development. (2) Is there a relationship between crystal behaviour and the turbulent under-ice boundary-layer structure? In particular it is useful to assess the role turbulence plays in suspending the crystals. (3) Finally, considering in broad terms what the large-scale implications of such finescale mechanics might be provides a way of contextualising the wider effect the observed conditions could have."

Rev2: L 89: 'over an ocean depth'

Au: change made

Rev2: L 103: Frazer et al. 2020 is in review.

**Au:** This is now published and the reference is amended - Frazer E.K., Langhorne P.J., Leonard G.H., Robinson N.J., Schumayer D.: Observations of the size distribution of frazil ice in an Ice Shelf Water plume, Geophys. Res. Lett., p.e2020GL090498 doi.org/10.1029/2020GL090498 2020.

**Rev2:** L 107: I suggest providing the dimensions of the ice hole.

Au: "1 m x 1 m x 2.3 m blocks of sea ice" now added

**Rev2:** L 110-111: Does this statement refer to platelet ice and frazil ice crystals? This needs clarification.

**Au:** The sentence specifically addresses where the crystal types fit within a size continuum. It now says "Hoppmann et al. (2020) describes the semantics of platelets and frazil. For present purposes we consider them to be ends of a spectrum of the same physical crystal evolutionary process."
**Rev2: L115: Is there any previous evidence that it is a tightly interlocked matrix?**

**Au:** Robinson et al. 2014 addresses this and they say... Loose accumulations of buoyant platelet ice crystals beneath sea ice in McMurdo Sound have previously been termed "subice platelet layers" [Paige, 1966; Gow et al., 1998; Crocker and Wadhams, 1989; Jones and Hill, 2001; Mahoney et al., 2011; Gough et al., 2012]. However, that term does not adequately describe the network of unconsolidated platelets observed in the present study in which individual crystals grew through each other after floating to the surface to create a delicate lattice. Hence, we use the term "subice platelet matrix" to describe a thick, porous, and semirigid framework that is bathed in a continuous supply of supercooled water sourced from under an ice shelf. The key word is "semirigid". We now include these references.

**Rev2:** L127-131: Where is the echosounder pointing? Upward to the SIPL base? Is the beam 3 m wide at 25 m depth or at the SIPL base? It would be very useful to have an estimation of the field of view.

**Au:** Good point as the original text was ambiguous. The transducer is facing downward and located right at the base of the SIPL. The text now says... "We placed a downward-looking Simrad EK60 200 kHz echo sounder right at the base of the SIPL recording acoustic backscatter at 1 Hz with 4 cm vertical resolution over a sampling cone that is 7 degrees across, so that at a depth of 25 m the cone is three m wide. The cone has side-lobes in the upper 5 m of the water column that are as wide as 30 deg. The beam-width is not particularly critical so long as it is wide-enough that scatterers register some rise component (vertical resolution). With horizontal flows of around 0.05 m s-1 this means that a reflector would stay in the beam for a maximum of nearly two minutes at 5 m with a 30-degree cone. Useful backscatter was consistently detected as deep as 30 m."

Rev2: L139: What is meant by superficial appearances? Can you develop and better
clarify this statement?

**Au:** We meant that if one quickly looked at the figure one might infer a horizontal dimension to the data that does not exist. We have re-worded. *"The angled trajectories relate to the crystal rise speed and are unconnected to the tidal advection. Horizon-tal flow controls the persistence of individual reflectors as it is responsible for moving crystals in and out of the acoustic beam."*

**Rev2:** L143-154: When were these measurements carried out and at what point on the tidal cycle?

**Au:** A number of CTD profiles were recorded but only one is used here. The timing of this is now described in Section 3.1.

**Rev2:** L 165-172: Again, when were these measurements carried out and at what point on the tidal cycle?

Au: As per above this has now been included in the text.

**Rev2:** L 175-176: What indications are there that this be the result of buoyantly forced ISW plume?

**Au:** We have moderated the claim and included some text and a reference. *"This is possibly the result of the buoyantly forced ice shelf water plume which is a consistent feature in the region (Robinson et al. 2014; Hughes et al. 2014) which, due to buoyancy, preferentially flows on the ice underside."*

**Rev2:** L 176-178: At what point on the tidal cycle were speeds of 0.1 m/s observed and in what direction? Please see comments provided below on Figure 3. It would be useful to provide date and month in parentheses after Julian Day.
Au: This is a good point to pick up on and there's a whole aspect of spring-vsneap mixing and the relationship to buoyancy induced currents. However, that is a different focus than the crystal rise vs turbulence aspect here. The neap phase has a strong non-tidal northward bias whereas moving into the spring tides it becomes more bi-directional. This "paradox" is common in estuarine conditions. We now add some text in the new section 4.3 where we connect what we have observed at the local scale to the wider system. It says.... "The timescale for this transport is of the order of a month. These data make it clear that the variation in tides over this period will influence the crystal sedimentation budget. First, the data indicate that the buoyancy driven upper ocean residual flow is faster and more continual during neap tides (Figure 3). So paradoxically, the low flow conditions are only seen away from neap conditions. The lowering of the flow speed and hence boundary layer turbulence if a shift to higher turbulent Péclet number conditions (Figure 11) so that buoyancy dominates. Consequently, the critical phase for the SIPL is slack water during spring tides which will allow crystals to settle out. The implication then is that the critical property to assess is the timescale over which a newly settled crystal becomes locked into the SIPL."

**Rev2:** L 180: I found the away and toward ice shelf convention confusing in the text and very difficult to interpret in Figure 3.

**Au:** We are not sure what more we can do here as the directions are in a "towards True North" angular convention and the ice shelves are to the south. We could put figure 3a b in terms of U and V components but we would still need to include a speed trace as this is the driver of the boundary-layer turbulence. We believe the additional text requested elsewhere by this Reviewer will emphasize why it is useful to have the speed.

Rev2: L 183: Is there a characteristic shape of platelet ice crystals?
**Au:** Good question – clearly they are flat and disk like. We have added in the distribution of minimum to maximum crystal dimensions to Fig. 4 showing the average ratio to be 0.67.

**Rev2:** L213-226: Can you be more specific about flow direction, point on the tidal cycle, temperature/salinity/backscatter changes and provide a date/time for each DOY.

Au: We originally included the arrows to highlight unambiguously the points we were discussing so the reader had absolute clarity about what we were talking about. Presumably the Reviewer is asking for these points to be carried over more directly into the text? We have amended the text slightly to carry across these points more clearly. "For most of the observation period the vertically integrated acoustic backscatter timeseries does not show any obvious consistent correlation with the velocity and scalar properties. However, exceptions occur near the end of day 298 (peak 2 in Figure 3) at which point we see shift from uni-directional flow through to a tidal oscillation and this comes at the end of a period of dropping temperatures (although only dropping by 30 mK) and increasing salinity. Simultaneously we observed the rise period to the highest backscatter (which is log-scale db). There is another instance where speed peaks correspond to changes in temperature, salinity and backscatter (peak 5). Near the end of the experiment at peak 7 again a velocity peak coincided with a change in backscatter, flow direction and salinity suggesting an entirely new water mass was moving by. It is noteworthy that the 3-4-day trends are comparable between temperature and acoustic backscatter as the pre day 299 conditions give way to a decline in backscatter while temperatures rise steadily over days 299-302. After day 302 the trends in both temperature and backscatter remain flat. These periods also correspond to changes in the tidal structure with unidirectional flow prior to day 298 then transitioning (days 299-302) to steady back and forth tides (post day 301)."

Rev2: L 230: What is the relevance of the frequency structure?

Au: We were not sure if this was a rhetorical question? Considering turbulence

TCD
through a frequency perspective is useful as it permits insight into the energy-baring scales and the turbulent cascade. This is especially true in a boundary layer where there might be topographic effects, convection vs shear etc. Being able to do this with the acoustics, velocity and scalar properties is really insightful. We have added an explanatory sentence and augmented the closing of the paragraph. *"Examination of the frequency structure of the forcing flow and responding timeseries (backscatter and temperature) provides clues as to the nature of the mechanics (Figure 8). The current meter energy spectrum is constrained to lower frequencies (n.b. the timeseries are not long enough to enable analysis to extend much lower than the diurnal frequency) with much of the spectrum above 50 cpd reaching an apparent noise-floor implying that the variations seen in temperature and backscatter are not advection-driven."*

**Rev2:** L 242: What is meant by 2-5 m undulation in SIPL underside? Where, when and how was this observed?

**Au:** This was observed throughout the experiment and we have now included a new figure panel (Fig 5b). We don't believe this evolved over the period of our experiment but being certain of this is future work.

Rev2: L 251: I suggest using profiles instead of realisations.

Au: Agreed and change made.

Rev2: L293: I suggest rewording the last sentence of this paragraph.

**Au:** This is a reasonable suggestion and the text now says... "The present situation is downstream of the Ross Ice Shelf cavity within which residence times in zero light likely to be in the range of 1-5 years (Reddy et al., 2010; Stevens et al., 2020) which further supports the suggestion that the targets and continuum are ice related rather than suspended biological organisms or sediment."
**Rev2:** L307-308: The Hoppmann et al. 2015 and Hunkeler et al. 2015 studies were not carried out in McMurdo Sound.

**Au:** Agreed, we did not mean to imply this but can see the references were poorly placed. We have removed the "at this location".

**Rev2:** L 380-405 and L 417-420: As above in the specific comments about the cooccurrence of the 'Dirty Ice' and SIPL distribution.

**Au:** AS per above we have removed the original subsection 4.3 and deleted or shifted the associated text.

Rev2: L424: Richter et al. 2020 is in review.

**Au:** We await an update on the Richter manuscript. We have replaced it with a reference by Colleoni et al. 2018.

**Rev2:** L424-432: This paragraph could be developed further to provide a more substantiated context and broader outlook for this study.

**Au:** Agreed. We have reworked and expanded this text. It now includes points relating to modelling scales as well as the interaction between tides, polynya and sea ice formation. *"Modelling studies focusing at the regional ISW plume scale require more observational data and context (Chen et al. 2019). Colleoni et al. (2018) show that ice shelf-sea ice-ocean connections remain a major outstanding challenge in models operating at climate timescales. Incorporation of the processes described here into modelling that typically resolve scales around 2 km or greater will be a challenge. A starting point might be the role present large platelets play in the McMurdo Sound polynya formation (Dai et al., 2020). Water column evolution and sea ice formation in polynya are spatially constrained phenomena driven by short, high-energy wind events that then drive formation of new sea ice. This energy conversion will be influenced*

TCD
by the presence of ISW that will enhance the processes. It is the by-product of this sequence that generates high salinity shelf water that ultimately has a global thermohaline impact. As large-scale models seek to improve their representation of polynya that will need to better account for the nature of ice shelf water plumes. Similarly, the present data suggest that such large spatiotemporal representations will need to find ways of including the effects of tidal fluctuations in some integrated fashion.

The topic is clearly still in a discovery phase with many fundamental questions remaining unanswered. This work suggests research themes for understanding sea ice formation near ice shelves should focus on the role of convection driven by SIPL crystal growth in modifying the turbulence in the upper water column and the feedbacks to the turbulence. In addition, the possible links between availability of nucleating material, crystal production and fate need to be examined, especially as to how this might support the formation of large, suspended ice crystals in an upper ocean influenced by tidal variations at both diurnal and spring-neap timescales."

Rev2: L 427: I do not understand this statement.

**Au:** Fair comment. As part of the response to the previous point we have re-worked the text to clarify the role these local-scale observations can play at the larger scale.

**Technical Comments:**

Rev2: 1) The Frazer et al. 2020 and Richter et al. 2020 studies are currently in review.

**Au:** The Frazer study has been published and we await an update on the Richter manuscript. We have replaced it with a reference by Colleoni et al. 2018.

**Specific Comments on Figures**

TCD
**Rev2:** Figure 1: I suggest combining a and c into one figure and masking out the land. A MODIS optical image showing the land-fast sea ice conditions at the time of the field campaign would be more informative as would stating the date/month of acquisition.

**Au:** We have developed a combined version of the figure. We left some aspects out to make it not too busy.

**Rev2:** Figure 2: At what point on the tidal cycle was this profile taken? And what are the DOY to relate to Figure 3?

**Au:** This has been amended and now says "This profile was from 0100 UTC on the 27th of October 2015 (DOY 300)". The associated text now says... "This example was recorded on DOY 300 as the tides moved into spring conditions and shows the upper layer..."

**Rev2:** Figure 3: I suggest moving a, b, c and d labels to top LH corner and put in parentheses; to make y-axis of tidal height larger. How was tidal height modelled? Please clarify convention for directions in a and b.

**Au:** The tidal amplitude is there to provide a spring/neap guide. We expand the tidal trace and moved the labels. Panel a is speed and has no direction and the panel b is direction towards True North as identified in the axis label and caption. Reference to the tidal height model is now included.

**Rev2:** Figure 6: State DOY to relate to Figure 3.

**Au:** The caption has been amended to now say... The segments come from (a) 25oct2015 at 1528 UTC (DOY 298.65), (b) 26oct2015 UTC at 0509 (DOY 299.84) and (c) 1nov2015 0438 UTC (DOY=305.19).

**Rev2:** Figure 12: Modify to account for changes to section 4.3 etc.

TCD
**Au:** Modified as requested... removed the wind-blown sediment, added in an arrow from the Ross cavity and modified the caption.

**Rev2:** Suggested reference: Cheng, C., Jenkins, A., Holland, P. R., Wang, Z., Liu, C., and Xia, R.: Responses of sub-ice platelet layer thickening rate and frazil-ice concentration to variations in ice-shelf water supercooling in McMurdo Sound, Antarctica, The Cryosphere, 13, 265–280, https://doi.org/10.5194/tc-13-265-2019, 2019.

**Au:** added with thanks and linked to and quoted in several places through the revised text.
Ross Sea fast ice McMurdo Sound **Ross Island** Ross Ice Shelf CA DIFC 🔴 HS CSS DI McMurdo Ice Shelf 20 km

Fig. 1. Revised Fig 1 - Location details...
TCD

Fig. 2. Fig 5 with new panel (b) relating to improved text.
TCD